# ✴ InfantAgent-Next: A Multimodal Generalist Agent for Automated Computer Interaction

**Bin Lei**[1][*] **Weitai Kang**[2][*] **Zijian Zhang**[1] **Winson Chen**[1] **Xi Xie**[3] **Shan Zuo**[3]
**Mimi Xie**[4] **Ali Payani**[5] **Mingyi Hong**[1] **Yan Yan**[2] **Caiwen Ding**[1]
[1]University of Minnesota  [2]University of Illinois Chicago
[3]University of Connecticut  [4]The University of Texas at San Antonio  [5]Cisco Research
lei00126@umn.edu, wkang126@uic.edu

## Abstract

This paper introduces INFANTAGENT-NEXT, a generalist agent capable of inter-acting with computers in a multimodal manner, encompassing text, images, audio, and video. Unlike existing approaches that either build intricate workflows around a single large model or only provide workflow modularity, our agent integrates tool-based and pure vision agents within a highly modular architecture, enabling different models to collaboratively solve decoupled tasks in a step-by-step manner. Our generality is demonstrated by our ability to evaluate not only pure vision-based real-world benchmarks (i.e., OSWorld), but also more general or tool-intensive benchmarks (e.g., GAIA and SWE-Bench). Specifically, we achieve a **7.27%** accuracy gain over Claude-Computer-Use on OSWorld. Codes and evaluation scripts are open-sourced at https://github.com/bin123apple/InfantAgent.

## 1 Introduction

Automated AI agents [28, 41, 35, 47, 29, 22] are vital in today's digital era. By integrating large language models (LLMs) [2, 19] and visual large language models (vLLMs) [36, 6, 32, 46, 10], they can multimodally understand user intents—including text, images, voice, and video—and, with minimal human intervention, convert these intents into precise sequences of interface actions. Through LLM / vLLM-driven planning, advanced perception of UI elements (such as buttons, text fields, and images), and a modular execution framework, they significantly reduce manual effort and enable the efficient execution of complex workflows across diverse software environments.

Current automated AI agents can be broadly classified into two categories, and they have not made sufficiently fine-grained distinctions at the agent's tool selection and execution levels, which leads to significant limitations in practice. ❶ The first category consists of *tool-based agents*, such as OpenHands [47], OWL [29], and AutoGPT [18], that equip LLMs with a suite of predefined tools (e.g., code generation tools, web search tools) to boost task-specific accuracy. However, because these agents usually rely on a single model to decide when and how to use each tool, they require the manual definition and integration of tools for every possible desktop scenario—an infeasible and brittle approach that limits generality. ❷ The second category comprises *pure vision-based agents*, such as UI-TARS [40], and Aguvis [54], which use vLLMs to control computers via GUI. This design allows broader applicability, since it bypasses the need for tool integration. However, high-resolution reasoning with a single model hurts accuracy on tasks that could be easily handled by simple tool calls, such as document editing or code manipulation—where tool-based agents excel. ❸ Although advanced models (e.g., Claude-3.7-Sonnet [6], o3 [38]) can decompose complex problems into precise, step-by-step plans, they frequently fail at execution—mislocalizing GUI click

---

[*]Equal Contribution.

coordinates [12] (GPT-4o [25] attains only $0.8\%$ accuracy on the ScreenSpot-Pro benchmark [30]) or selecting incorrect line numbers during file edits. Conversely, specialized vision modules (such as visual-grounding models [31, 17]) exhibit limited reasoning capacity and restricted context windows, preventing reliable inference of subsequent actions. Ensuring both high task-level accuracy and broad generality requires a hybrid agent paradigm that unifies tool-based and pure-vision approaches.

Inspired by these challenges, we present INFANTAGENT-NEXT, which undertakes detailed modularization of agent workflows, tool selection, and tool execution, in favor of a modular architecture with a unified dialogue context. Each subtask is routed to the most appropriate specialist—reasoning models handle logical inference, visual-grounding models localize UI elements, audio-analysis models interpret sound, and so on—and their outputs are seamlessly merged back into the conversation history. This design enables true multimodal interaction with computer interfaces, rather than being confined to formatted HTML or Accessibility Tree manipulations or purely vision-based control. Figure 1 presents several real-world task examples addressed by INFANTAGENT-NEXT.

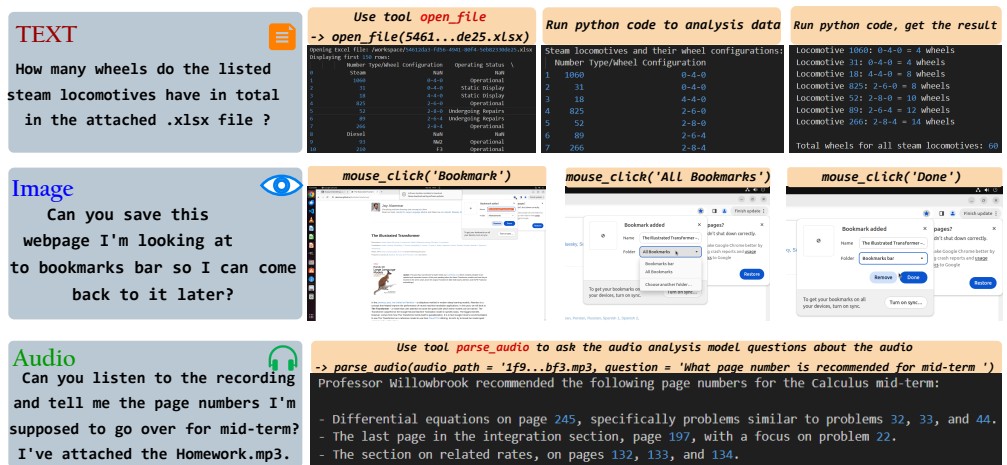

Figure 1: Three real-world task examples addressed by INFANTAGENT-NEXT, each requiring different modality capabilities. Input requests are shown in the ▇ block, actions taken by the agent are shown in the ▇ block, and execution results appear directly below the action block.

Our contributions are as follows: **(i)** We introduce INFANTAGENT-NEXT, an open-source, multimodal generalist agent framework that interacts with its host computer via text, images, and audio. **(ii)** We have modularized the agent's workflow, tool selection, and tool execution in detail—unifying the tool-based and pure-vision agent paradigms to achieve both high task-level accuracy and broad generality. **(iii)** We optimize a suite of commonly used agent tools and release them as open-source which supports isolated execution environment, enabling future research in the community.

## 2 Related Work Comparison

In Table 1, we compare INFANTAGENT-NEXT with existing agents across eight key dimensions.

***Generalist:*** Generalist agents are capable of supporting various computer tasks, while specialist agents are typically designed to address a specific class of problems. For example, OpenHands [47] and MetaGPT [22] primarily focus on software engineering tasks.

***Built-in Tools:*** For common operations such as web search or file editing, integrating built-in tools into agents can greatly streamline workflows and enable structured output generation. For instance, AutoGPT [18]'s `search_web` tool retrieves search results by issuing HTTP requests and parsing the returned JSON/HTML structure. In contrast, agents like UI-TARS [40] lack such tools and rely solely on pure vision-based capabilities for parsing.

***Visual Grounding:*** Built-in tools are often insufficient for diverse computer softwares—particularly with professional graphics software or niche PC games. Agents such as ShowUI [31], which are equipped with visual grounding capabilities, are better suited for handling these scenarios.

*Multi-Model Support:* Different models exhibit different strengths: reasoning models are effective at complex analytical tasks, non-reasoning models offer faster response times, audio analysis models handle auditory inputs, and visual grounding models enable precise spatial localization. Assigning each model to tasks that match its strengths can significantly enhance overall agent performance. MetaGPT [22] exemplifies this multi-model integration strategy.

*Memory Retrieval:* As dialogue histories grow, model inference overhead increases. This overhead can be mitigated by retrieving relevant memory segments and injecting them into the context on demand. LangChain [27] demonstrates this approach by offering multiple memory classes—such as `summary memory` and `retrieval-based memory`.

*Dynamic Toolset:* To extend an agent's functionality, developers often include a broad range of tools. Tool usage instructions are typically prepended to the conversation history to guide the LLM. However, a large tool inventory can lead to increased inference cost and challenging tool selection. Frameworks like LangChain [27] address this by dynamically selecting a relevant subset of tools from the full library, thereby reducing the cognitive and computational burden.

*User Interaction during Execution:* User's initial prompts are sometimes underspecified (e.g., *Please write a short story.*). In such cases, it is crucial for the agent to interactively elicit additional details rather than completing the task blindly. For example, OpenHands [47] includes a `MessageAction` class that enables the agent to request clarification from the user.

*Dedicated Computer:* We design a fully isolated execution environment supporting CLI, Python scripting, and GUI interaction; implementation details are provided in the Appendix D.

Table 1: Comparison of INFANTAGENT-NEXT with Different Agent Frameworks

| Name | Generalist Agent | Built-in Tools | Visual Grounding | Multi-Model Support | Memory Retrieval | Dynamic Toolset | User Interaction during Execution | Dedicated Computer |
|---|---|---|---|---|---|---|---|---|
| AutoGPT [18] | ✓ | ✓ | ✗ | ✓ | ✗ | ✗ | ✗ | ✗ |
| BabyAGI [34] | ✓ | ✗ | ✗ | ✗ | ✗ | ✗ | ✗ | ✗ |
| LangChain [27] | ✓ | ✓ | ✗ | ✓ | ✓ | ✓ | ✓ | ✗ |
| MetaGPT [22] | ✗ | ✓ | ✗ | ✓ | ✗ | ✗ | ✗ | ✗ |
| OpenHands [47] | ✗ | ✓ | ✗ | ✓ | ✗ | ✗ | ✓ | ✗ |
| UI-TARS [40] | ✗ | ✗ | ✓ | ✗ | ✗ | ✗ | ✗ | ✗ |
| AutoGen [49] | ✓ | ✓ | ✗ | ✓ | ✗ | ✗ | ✗ | ✗ |
| AGUVIS [54] | ✗ | ✗ | ✓ | ✗ | ✗ | ✗ | ✗ | ✗ |
| ShowUI [31] | ✗ | ✗ | ✓ | ✗ | ✗ | ✗ | ✗ | ✗ |
| OS-Atlas [51] | ✗ | ✗ | ✓ | ✓ | ✗ | ✗ | ✗ | ✗ |
| AutoAgents [11] | ✓ | ✗ | ✗ | ✓ | ✗ | ✓ | ✗ | ✗ |
| **InfantAgent-Next** | ✓ | ✓ | ✓ | ✓ | ✓ | ✓ | ✓ | ✓ |

Compared to prior agent frameworks, INFANTAGENT-NEXT uniquely integrates all of the above features. See Section 3 for further details.

## 3 InfantAgent-Next

In this section, we first introduce the overall architecture of INFANTAGENT-NEXT to illustrate its operational logic in Sec. 3.1. We then describe some crucial components in Sec. 3.2, including the memory extraction mechanism employed during execution, toolkit configured for the system, etc. Finally, we propose our novel improvement on important actions of the agent in Sec. 3.3, including Mouse Click and File Edit. We provide a detailed case study in Appendix A.

### 3.1 Architecture

*Overview:* The architecture of INFANTAGENT-NEXT is illustrated in Figure 2. INFANTAGENT-NEXT meticulously modularizes the workflow where different models complete different steps. The process begins when the user submits a request. We first allow the user to configure the arguments for Workflow models and Tool models. After the agent is initialized, it analyzes the user's request to generate tasks. It then performs tool selection and invokes the selected tool to complete the tasks. Finally, the agent checks whether the current task has been successfully completed.

**I**. *Argument Configuration* (Figure 2-upper left): In this stage, users can customize and assign different models to different tasks. These include three workflow models responsible for planning, tool selection, and task execution, respectively, and three modality-specific tool models for handling images, audio, and video.

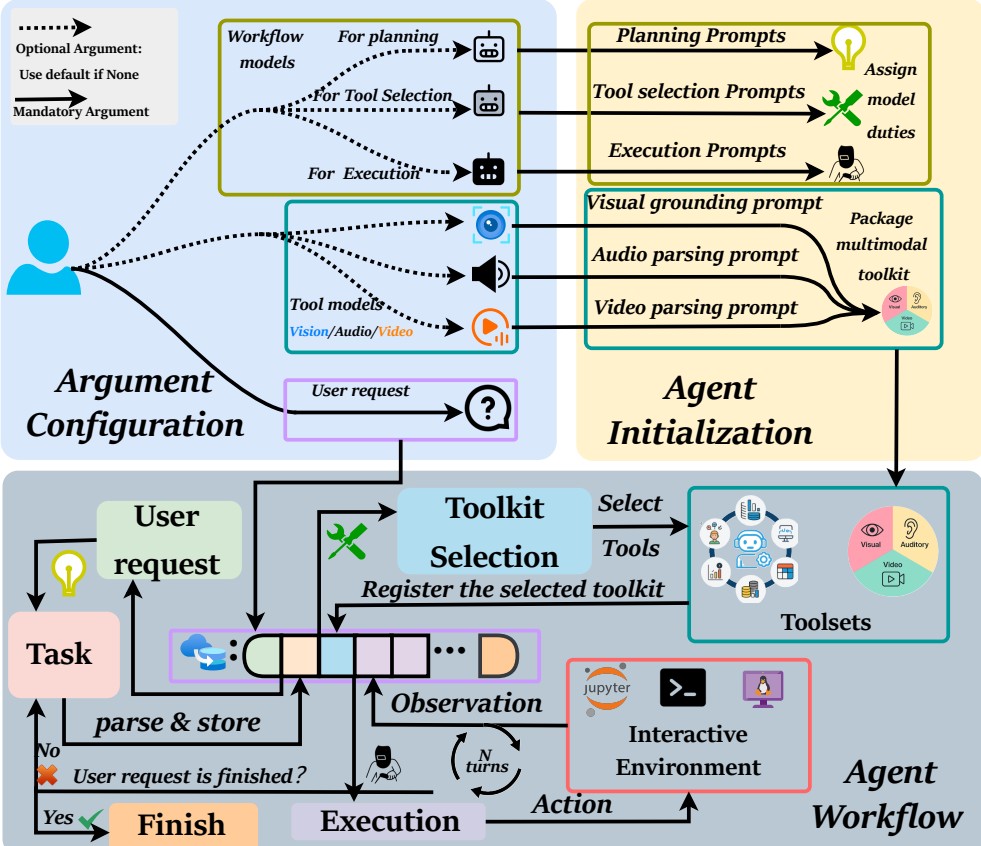

Figure 2: INFANTAGENT-NEXT architecture overview. 🔵: User input argument and request. **Environment related icons:** □: Agent Interaction Environment. ▶: Terminal interface. 🖥: GNOME desktop. 🪐: Jupyter. **Models related icons:** ☐: Load Workflow models. 🤖: Planning model 🤖: Tool Selection model 🤖: Execution model 💡: Planner. ✖: Tool Selector. 👤: Executer. **Tools related icon:** □: Load Toolsets. 🔵: Tool models argument: `Vision_model_name`. 🔊: Tool models argument: `Audio_model_name`. 🔵: Tool models argument: `Video_model_name`. 🎨: Multimodal toolkit. ⚙: Other Toolkits. **Memory related icons:** □: Load agent memory. ②: Request from the user. 🔵: Agent Memory Cache, used for storing all memories.

**II**. *Agent Initialization* (Figure 2-upper right): During initialization, we assign specific roles to each model with tailored prompt templates. For example, some tools are associated with specific tool models and their prompt templates, and are then packaged into a unified multimodal toolkit. At this stage, the agent's memory cache is initialized, with the user's request stored as the first entry.

**III**. *Agent Workflow* (Figure 2-bottom): The initialized multimodal toolkits, together with a suite of predefined utility functions, are organized into a comprehensive toolset. This set of tools, along with the interactive environment, all models, and the agent's memory cache, is integrated into the agent's workflow. The user's initial request serves as the input and triggers the following iterative loop: ❶ The Planner model analyzes the request and current state to produce a task, which is parsed and stored in memory. ❷ The agent reconstructs the memory using a prompt template for toolkit selection. The Tool Selection model then selects the most appropriate tool from the toolset. The selected tool is registered and activated for this iteration, while others remain idle. ❸ The Executor model invokes the tools and gathers feedback from the environment to complete the task. ❹ If we successfully resolve the task, the loop terminates; otherwise, the process proceeds to the next iteration.

## 3.2 Component Design

*Memory:* Since the agent must perform three distinct types of tasks—planning, tool selection, and execution—it is insufficient to simply log the raw dialogue history. Instead, each model-

generated response is parsed using special tags and sequentially stored in the agent's memory cache. The agent's reasoning is enclosed within `<thought>...</thought>` tags; task assignments are marked by `<task>...</task>`; selected toolkits are indicated by `<toolkit>...</toolkit>`; and tool executions are represented by either `<execute_bash>...</execute_bash>` or `<execute_python>...</execute_python>`. When the agent transitions to a new task, a subset of the memory cache is extracted and reconstructed into the dialogue by concatenating relevant memory attributes. Task-specific tags are activated to prevent the LLM from generating irrelevant content. Finally, the reconstructed dialogue is combined with predefined prompt templates to form the conversation history for that task. The restoration process follows these steps:

**I**. *Planning*: During the planning phase, the agent retrieves all memories except those associated with tool selection. All tags are removed during reconstruction, except for the `<task>...</task>` tags. These reconstructed memories are inserted between the planning system prompt and the end prompt for planning, both of which are detailed in Appendix B.1.

**II**. *Tool selection*: For tool selection, only the most recent task memory is retrieved from the cache, and all associated tags are removed. This content is then appended directly after the tool selection system prompt, as provided in Appendix B.2.

**III**. *Execution*: During execution, all memories except those related to tool selection are retained. When reconstructing them into dialogue, only the `<execute_bash>...</execute_bash>` and `<execute_python>...</execute_python>` tags are preserved. The resulting dialogue is inserted between the execution system prompt and execution end prompt, shown in Appendix B.3.

***Tools:*** To invoke a tool, the agent prepends its documentation and usage examples to the dialogue history. However, as the toolset expands, this strategy introduces two challenges: (1) inflated inference overhead due to the volume of text, and (2) increased difficulty for the model to identify the correct tool. To mitigate these issues, we categorize tools and dynamically select a relevant subset during each tool selection step. Specifically, we define seven toolkits: File Reading, File Searching, File Editing, Web Browsing, Computer Use, Code Execution, and Advanced Tools. The Computer Use toolkit includes all keyboard and mouse operations as well as multimodal tools, while the Advanced Tools toolkit provides various composite commands. Each toolkit is accompanied by a usage example. Detailed descriptions of tool functions are available in Appendix C.

### 3.3 Mouse Click & File Edit

As noted earlier, SOTA models, such as Claude-3.7-Sonnet or GPT-4o, remain limited in mouse-click action (visual grounding) accuracy and robust file editing. We incorporate dedicated mechanisms to mitigate these limitations.

***Mouse Click:*** For `mouse_click` operations, the model is required to output two parameters: (1) the name of the target element (e.g., Google Chrome or VS Code icon), and (2) a detailed description of its position and shape. As illustrated in Algorithm 1, we first ver-

---
**Algorithm 1** Iterative Region Cropping and Mouse Click Logic

**Require:** *agent*
1: **if** Last(*agent.memory*).type $\neq$ "mouse_click" **then**
2:      **return**
3: **end if**
4: $description \leftarrow$ Last(*agent.memory*)$.description$
5: $region \leftarrow screenshot$
6: **for** $i = 1$ **to** $n$ **do**
7:      $coord_i \leftarrow$ VISIONGROUNDING($description, region$)
8:      $region \leftarrow$ CROPSCREEN($center = coord_i$, $size =$ CROP_SIZE)
9: **end for**
10: $final\_coord \leftarrow$ VISIONGROUNDING($description$, $region$)
11: PERFORMCLICK(final_coord)

---

ify whether the agent's most recent action is `mouse_click`; if not, the procedure terminates early. Otherwise, we retrieve the associated description from memory and treat a full-screen screenshot as the initial search region. We then enter an iterative loop of length $n$, where each iteration passes the current region and the description to the visual grounding model to obtain a candidate coordinate $coord_i$. We subsequently call `CropScreen` centered at $coord_i$ to extract a smaller region for the next step. After completing $n$ iterations, the final region is passed to the visual grounding model once more to predict the refined click position $final\_coord$, which is then used by `PerformClick`

to click on the screen. By progressively narrowing the search region, this method enhances visual grounding precision and reduces distraction from irrelevant pixels.

*File Editing:* There are two main formats for the `file_edit` command. (1) The three-parameter format used by SWE-Agent [44] specifies the file path, the exact string to replace, and the replacement. It often fails when the target string is long or occurs multiple times. (2) The four-parameter format used by OpenHands specifies the file path, start and end line numbers, and the replacement string, but is prone to errors such as incorrect or overlapping line ranges. To ensure that the agent can perform file edits accurately, we have encapsulated the logic from Algorithm 2 as follows: ❶ First, we verify whether the agent's most recent memory entry is of type `file_edit`. If not, the procedure terminates immediately. ❷ If it is, the agent extracts the edit request from memory and invokes `GenerateEditPlan(request)`, which returns the proposed starting and ending line num-

---

**Algorithm 2** File Editing Logic

---

**Require:** *agent*
1: **if** Last(*agent.memory*).type $\neq$ "file_edit" **then**
2:     **return**
3: **end if**
4: $request \leftarrow$ Last(*agent.memory*).edit_details
5: **for** $i \leftarrow 1$ **to** MAX_ITER **do**
    `#Generate edit plan`
6:     $(s, s_{\text{cont}}, e, e_{\text{cont}}) \leftarrow$ GENERATEEDITPLAN$(request)$
    `#Verify plan boundaries`
7:     **if** FileLine$(s) = s_{\text{cont}} \wedge$ FileLine$(e) = e_{\text{cont}}$ **then**
8:         APPLYEDIT(s, e, request)
9:         **return**
10:     **end if**
    `#Fallback:  locate best matching lines`
11:     $a_s \leftarrow$ FileLine$(s)$
12:     $a_e \leftarrow$ FileLine$(e)$
13:     $m_s \leftarrow$ FUZZIFY$(s_{\text{cont}})$
14:     $m_e \leftarrow$ FUZZIFY$(e_{\text{cont}})$
15:     $(s', e') \leftarrow$ FINDBESTMATCH$(m_s, m_e)$
16:     $request \leftarrow (s', e', a_s, a_e, request)$
17: **end for**

---

bers $s$ and $e$, along with the expected contents of those lines, *s_cont* and *e_cont*. ❸ The agent then checks whether the actual content of line $s$ matches *s_cont*, and whether line $e$ matches *e_cont*. If both checks pass, it directly applies the edit using `ApplyEdit(s, e, request)` and terminates. ❹ If the boundary checks fail, the agent switches to a fallback strategy: it first saves the original contents at lines $s$ and $e$ as *a_s* and *a_e*, then fuzzifies the expected snippets via `m_s = Fuzzify(s_cont)`, `m_e = Fuzzify(e_cont)` and calls `FindBestMatch(m_s, m_e)` to locate the most similar span $(s', e')$ in the file. The edit request is then updated with the new span and original content $(s', e', a\_s, a\_e, request)$, and the process repeats for up to `MAX_ITER` iterations.

## 4 Experiment

### 4.1 Vision Ability

*Benchmark.* We assess the visual reasoning capability of INFANTAGENT-NEXT using the OSWorld benchmark [53]. OSWorld provides a scalable, real-world computer environment comprising 369 open-ended desktop tasks. Each task is initialized with a fully specified machine state, including a high-resolution screenshot, active application windows, and file system context, paired with a natural language instruction and an executable evaluation script. The tasks span diverse domains, such as web browsing, file operations, code editing, image manipulation, and multi-application workflows, challenging the agent's ability to ground natural language instructions in GUI elements.

*Experiment Setup.* The evaluation is conducted using the official OSWorld codebase [53], executed within a VMware virtual machine. Mouse click events are simulated using `PyAutoGUI`. For reasoning, we use `Claude-3.7-Sonnet`, while visual grounding is performed using `UI-TARS-1.5-7B`, with `max_steps` set to 50. For fair comparison with baseline methods, all non-visual toolkits are disabled.

*Results.* As presented in Table 2, INFANTAGENT-NEXT achieves superior accuracy at 50 steps compared to OpenAI CUA and Claude Computer Use. When utilizing `Claude-3.7-Sonnet` as the planner model, INFANTAGENT-NEXT outperforms Agent-S2 [3] and all other open-source frameworks, demonstrating its enhanced visual reasoning capabilities.

Table 2: Performance of INFANTAGENT-NEXT on OSWorld. 🔒: Close source. ✓: Open source.

| Framework | Model | Steps | Open Source | Accuracy |
|---|---|---|---|---|
| UI-TARS [40] | UI-TARS-1.5-72B | 100 | 🔒 | 42.5 |
| OpenAI CUA [37] | GPT-4o | 200 | 🔒 | 38.1 |
| UI-TARS [40] | UI-TARS-1.5-72B | 50 | 🔒 | 38.0 |
| InfantAgent-Next | Claude-3.7-Sonnet + UI-TARS-1.5-7B | 50 | ✓ | 35.3 |
| Agent S2 [3] | Claude-3.7-Sonnet | 50 | ✓ | 34.5 |
| OpenAI CUA [37] | GPT-4o | 50 | 🔒 | 32.6 |
| Claude Computer Use [7] | Claude-3.7-Sonnet | 100 | 🔒 | 28.0 |
| UI-TARS [40] | UI-TARS-1.5 7B | 100 | ✓ | 26.9 |
| Claude Computer Use [7] | Claude-3.7-Sonnet | 50 | 🔒 | 26.0 |
| UI-TARS [40] | UI-TARS-72B-DPO | 50 | ✓ | 24.6 |
| AGUVIS [54] | AGUVIS 72B + GPT-4o | - | ✓ | 17.0 |
| Aria-UI [55] | Aria-UI + GPT-4o | - | ✓ | 15.1 |
| OS-Atlas [51] | OS-Atlas-Base-7B + GPT-4o | - | ✓ | 14.6 |
| SeeClick [12] | SeeClick + GPT-4o | - | ✓ | 9.21 |
| Qwen2.5 [10] | Qwen2.5-vl-72B | - | ✓ | 8.8 |

Table 3: Performance on SWE-Bench-Verified and SWE-Bench-Lite.

| Open/Close Source | Method | Accuracy (%) |
|---|---|---|
| **SWE-Bench-Verified (50 cases)** | | |
| | SWE-agent + Claude 3.7 Sonnet w/ Review Heavy [44] | 72 |
| | Openhands_04_15 [47] | 68 |
| | InfantAgent-Next + Claude-3.7-Sonnet | 66 |
| Open Source | AutoCodeRover-v2.0 (Claude-3.5-Sonnet-20241022) [9] | 52 |
| | SWE-agent + SWE-agent-LM-32B [44] | 46 |
| | AppMap Navie v2 [8] | 12 |
| | Agentless Lite + O3 Mini (20250214) [52] | 10 |
| | CodeStory Midwit Agent + swe-search [4] | 70 |
| | AgentScope [15] | 66 |
| Close Source | CORTEXA [1] | 62 |
| | Amazon Q Developer Agent_2024_12_02 [5] | 54 |
| | devlo_2024_11_08 [13] | 48 |
| **SWE-Bench-Lite (using GPT-4o)** | | |
| | Agentless-1.5 [52] | 32.00 |
| Open Source | InfantAgent-Next | 31.67 |
| | Agentless + RepoGraph [52] | 29.67 |
| | OpenHands + CodeAct v1.8 [47] | 22.00 |
| Close Source | CodeShellTester [48] | 31.33 |
| | SIMA [43] | 27.67 |

## 4.2 Logic Reasoning Ability

***Benchmark.*** To rigorously evaluate the logical reasoning capabilities of INFANTAGENT-NEXT, we employ two complementary SWE-Bench [26] datasets. First, SWE-Bench-Lite, a 300-issue subset, challenges the agent to interpret real GitHub bug reports, identify defects in a Python codebase, propose patches, and verify fixes by executing the provided test suites. This setup emphasizes multi-step logical planning rather than GUI proficiency. Second, we assess performance on the full SWE-Bench-Verified suite (500 cases); to manage API-related costs, we uniformly sample 50 cases for evaluation (their instance IDs are listed in the Appendix E). Together, these benchmarks provide a thorough yet cost-effective assessment of our agent's reasoning capabilities.

***Experiment Setup.*** For SWE-Bench-Lite, our inference is conducted using GPT-4o for planning, tool selection, and execution, consistent with the configurations adopted by most existing agents to ensure fair comparisons. For the 50-case SWE-Bench-Verified subset, we employ `Claude-3.7-Sonnet` for each step, providing direct evaluation against state-of-the-art systems. In both settings, `DeepSeek-V3-0324` is used as the dedicated file-editing model.

Table 4: Performance of INFANTAGENT-NEXT on GAIA. The data compared in this table are current as of the completion date of our GAIA experiments on April 17, 2025. For results without citations, please refer to the official GAIA leaderboard [14].

| Open/Close Source | Agent Name | Main Model | Average | Level 1 | Level 2 | Level 3 |
|---|---|---|---|---|---|---|
| Close Source | Langfun Agent v2.1 [16] | Claude-3.7-sonnet | 71.52 | 83.02 | 68.60 | 57.69 |
| | Trase Agent v0.3 [45] | o1 | 70.30 | 83.02 | 69.77 | 46.15 |
| | h2oGPTe Agent v1.6.8 [20] | Claude-3.5-Sonnet | 63.64 | 67.92 | 67.44 | 42.31 |
| | Anges | - | 60.00 | 66.04 | 65.12 | 30.77 |
| | desearch | GPT-4o | 56.97 | 71.70 | 58.14 | 23.08 |
| | Ormind v0.1 [39] | - | 55.15 | 69.81 | 54.65 | 26.92 |
| | Barcelona v0.1 | Claude-3.5-sonnet | 50.30 | 62.26 | 50.00 | 26.92 |
| | DRP-val-v.1.0 | - | 46.06 | 56.60 | 48.84 | 15.38 |
| | omne | o1-preview | 46.06 | 60.38 | 44.19 | 23.08 |
| | qbnlp | o1 | 44.24 | 52.83 | 45.35 | 23.08 |
| | LRC-Huawei | - | 40.61 | 52.83 | 43.02 | 7.69 |
| | Agent-On-the-fly | QwQ-32B | 40.61 | 52.83 | 43.02 | 7.69 |
| | AgentIM v1.1 | gpt-4-turbo | 40.00 | 50.94 | 40.70 | 15.38 |
| Open Source | OWL-Roleplaying | GPT-4o + o3-mini | **58.18** | **81.13** | 54.65 | 23.08 |
| | InfantAgent-Next | Claude-3.7-Sonnet | 56.97 | 62.26 | **62.79** | 26.92 |
| | TapeAgents [42] | Claude-3.7-Sonnet | 55.76 | 71.70 | 53.49 | **30.77** |
| | Auto-Deep-Research [21] | Claude-3.7-Sonnet | 55.15 | 71.70 | 53.49 | 26.92 |
| | AutoAgent | - | 55.15 | 71.70 | 53.49 | 30.77 |
| | Open Deep Research [24] | o1 | 55.15 | 67.92 | 53.49 | 34.62 |
| | Magnetic-1 | o1 | 46.06 | 56.60 | 46.51 | 23.08 |
| | HuggingFace Agents [23] | gpt-4o | 44.24 | 58.49 | 43.02 | 19.23 |
| | Multi-Agent Exp v0.1 | gpt-4-turbo | 39.39 | 54.72 | 38.37 | 11.54 |
| | FRIDAY [50] | gpt-4-turbo | 34.55 | 45.28 | 34.88 | 11.54 |

***Results.*** On the SWE-Bench-Verified benchmark (50 cases), INFANTAGENT-NEXT achieves a leading accuracy of **66%**, outperforming many proprietary agents. Notably, several closed-source agents such as Amazon Q Developer Agent and Emergent E1 perform worse, highlighting the effectiveness of our agent's architecture and integration. On the SWE-Bench-Lite benchmark, where we ensure a fair comparison by restricting all agents to use GPT-4o, INFANTAGENT-NEXT maintains competitive performance with an accuracy of **31.67%**. This places our system among the top-performing open-source agents, demonstrating its robustness and adaptability.

## 4.3 General Task Performance

***Benchmark.*** We evaluate INFANTAGENT-NEXT on the GAIA benchmark [33], designed to assess general AI assistants. GAIA comprises open-ended, real-world questions across three difficulty tiers—Level 1 (basic), Level 2 (intermediate), and Level 3 (advanced). These tasks require integrating core capabilities, including reasoning, multimodal understanding, web navigation, and tool usage, to produce a single, verifiable answer.

***Experiment Setup.*** The evaluation is conducted on the GAIA benchmark validation set. We employ `Claude-3.7-Sonnet` for reasoning and task execution, `Deepseek-V3-0324` for tool selection, `gpt-4o-audio-preview` for auditory processing, and `UI-TARS-1.5-7B` [40] for visual understanding. System prompts are sourced directly from the official GAIA leaderboard to ensure consistency with standard evaluation protocols.

***Results.*** The performance on the GAIA benchmark is presented in Table 4. While closed-source agents maintain a performance edge, INFANTAGENT-NEXT ranks second among open-source agents, trailing only OWL, and achieves state-of-the-art results on Level 2 difficulty questions.

## 4.4 Ablation on Visual Grounding design

***Benchmark.*** ScreenSpot-Pro [30] provides a high-resolution professional scenarios to stress-test visual grounding. It contains 1,581 annotated screenshots drawn from 23 applications spanning five industry domains and three operating systems; each sample pairs a natural-language prompt with the exact on-screen target element, enabling precise click-accuracy evaluation. We therefore use ScreenSpot-Pro to quantify the performance of our newly designed Iterative Region Cropping.

***Experiment Setup.*** We performed our experiment using 2x A100 80G GPU. We first conduct an ablation study on the Iterative Region Cropping setup. Specifically, for `CROP_SIZE`, we test three

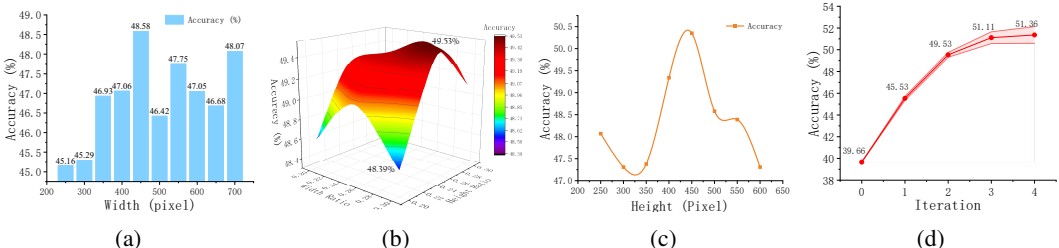

(a)  (b)  (c)  (d)

Figure 3: We conduct an ablation study on the Iterative Region Cropping setup from four perspectives. (a) Vary the region width. (b) Vary the width and height ratios of the cropped region (relative to the full image) (c) Vary the height. (d) Vary the number of iteration

different settings with a fixed iteration length of $n = 2$ to evaluate its impact: (Fig. 3 (a)) we fix the cropped region area to 150K pixels and vary the region width. (Fig. 3 (b)) we examine how varying the width and height ratios of the cropped region (relative to the full image) affects performance. (Fig. 3 (c)) we fix the width at 700 pixels and vary the height. (Fig. 3 (d)) we further ablate the number of iterations, fixing the region width and height ratios to 0.3 and 0.25, respectively.

**Results.** Among all strategies, the setting in Fig.3 (c)—fixing the width while varying the height—yields the highest accuracy and is preferred for determining CROP_SIZE. For Fig. 3 (d), accuracy converges at the third iteration, and using two iterations offers a favorable trade-off between performance and inference cost.

## 4.5 Evaluation on File Editing

**Benchmark.** We evaluate the file editing capabilities of INFANTAGENT-NEXT using the SWE-Bench-Verified dataset [26], a curated subset of 500 real-world GitHub Issue–Pull Request pairs from popular Python projects. Each task is human-validated to ensure sufficient context and is verifiable through the project's unit tests. Tasks involve diverse code modification instructions, such as deleting or updating code across multiple files, testing the agent's ability to interpret and implement complex changes. We randomly sampled 10% of the tasks from the SWE-Bench-Verified dataset across repository categories to form our evaluation set. The IDs of the selected test cases are in the Appendix E.

**Experiment Setup.** We evaluate each task solution and categorize them into three sets: $C_{\text{total}}$, the set of all task solutions; $C_{\text{fail}}$, solutions failing due to execution-model errors; $C_{\text{repaired}}$, solutions successfully repaired. We define the following metrics: $\text{RepairSuccessRate (RSR)} = |C_{\text{repaired}}|/|C_{\text{fail}}|$ and $\text{OverallRepairRate (ORR)} = |C_{\text{repaired}}|/|C_{\text{total}}|$

**Results.** INFANTAGENT-NEXT exhibits strong performance on file editing tasks in the SWE-Bench-Verified dataset, achieving a total success rate of 90.4% with 84.3% RSR and 51.4% ORR. The agent excels at fixing line number off-by-one errors (30% repaired), content unmatched (13% repaired), and Python syntax errors (6% repaired), demonstrating robust code interpretation and repair capabilities. Overall, these results highlight INFANTAGENT-NEXT as a highly capable agent for automated file editing, with minor limitations in edge cases.

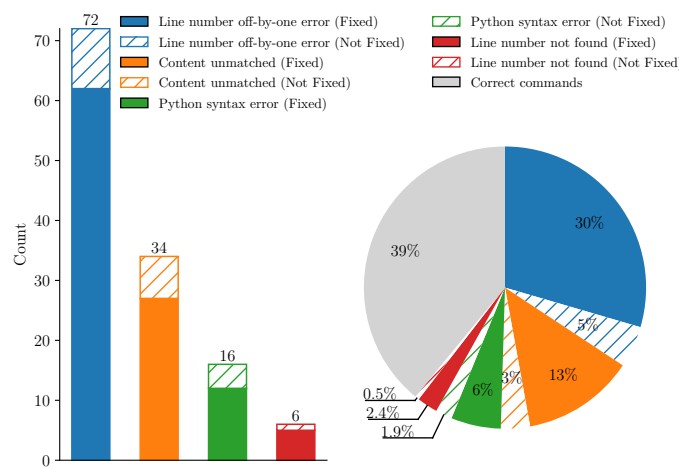

Figure 4: Evaluation on a subset of SWE-Bench-Verified.

# 5 Conclusion

We presented INFANTAGENT-NEXT, a multimodal generalist agent that bridges the strengths of tool-based and pure vision-based paradigms through a modular, context-aware architecture. By routing subtasks to specialized models and maintaining a unified dialogue context, INFANTAGENT-NEXT avoids the limitations of single-model systems and delivers both high task accuracy and broad applicability across diverse interfaces. Empirical results on OSWorld, GAIA, and SWE-Bench confirm its effectiveness, including a **7.27**% accuracy gain over Claude-Computer-Use. All code, models, and evaluation tools will be released to support future research in multimodal agent design.

## Acknowledgment

This research was partially supported by Cisco Research. We thank the open-source community for their valuable suggestions, issues, and pull requests.

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

# A    Case Analysis

Figure 5 illustrates the step-by-step process by which INFANTAGENT-NEXT solves a real-world query: *"According to the World Bank, which countries had gross savings of over 35% of GDP for every year in the period 2001–2010?"*

To answer this question, INFANTAGENT-NEXT performs three sequential cycles of *Planning → Tool Selection → Execution*:

**First loop:** The agent selects the `Web_Browser` toolkit, conducts a web search, performs mouse-click operations, and captures desktop screenshots as execution outputs. This enables it to successfully locate and download the relevant CSV file.

**Second loop:** It then selects the `File_Search` and `File_View` toolkits, using Bash commands to inspect the directory structure and examine the contents of the CSV file.

**Third loop:** Finally, the agent selects the `Code_Execution` toolkit and executes Python code to parse and aggregate the data in order to identify qualifying countries.

Through these modular and interpretable steps, INFANTAGENT-NEXT arrives at the correct answer with minimal human intervention.

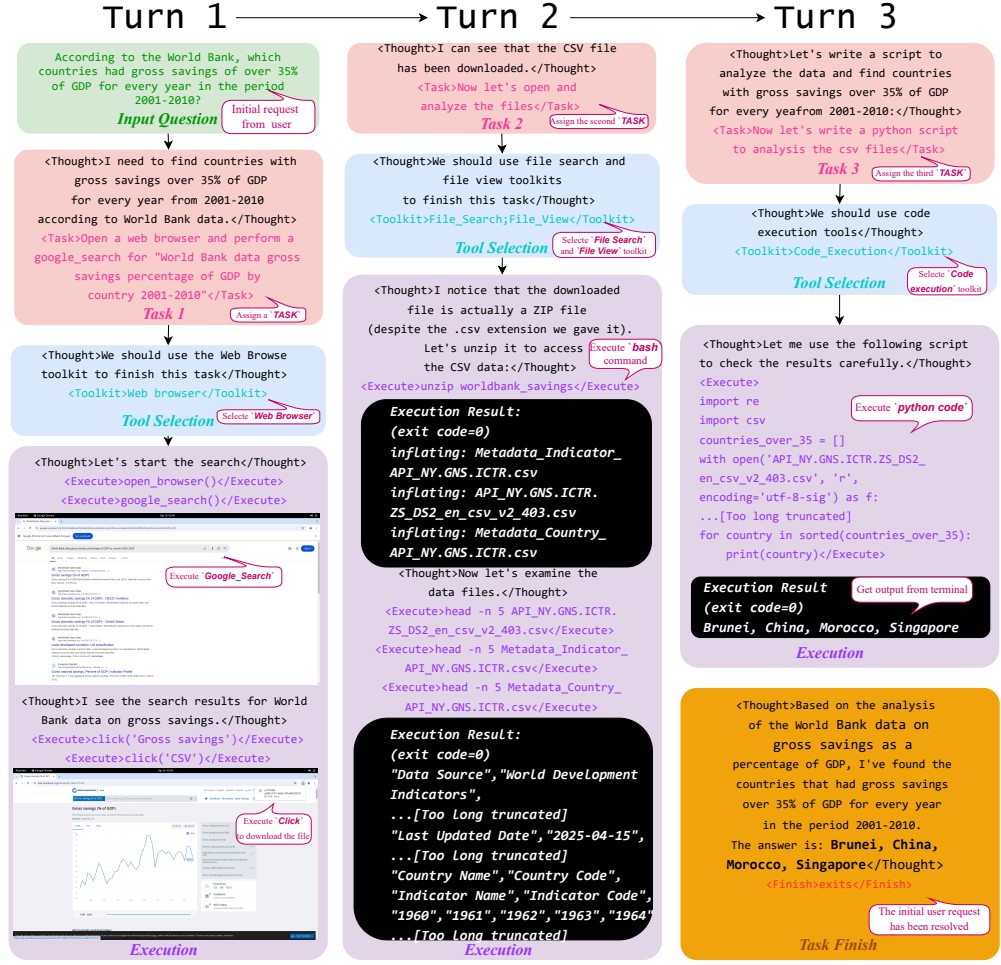

Figure 5: Cases analysis. Zoom in to view the detailed content in the screenshot.

# B    Workflow Prompt Templates

## B.1    Planning

Listing 1: Planning system prompt. It is inserted at the beginning of each planning.

```
In our interaction with the **User (requester)**, our goal is to
    gradually resolve their request or answer their questions.
The process involves three roles: **You (reasoner)**, **Me (executor)
    **, and **User (requester)**.
In each response, you must execute only one step, choosing to either
    assign a task to **Me (executor)**, or request information from
    the **User (requester)**.
**I (executor)** can help **You (reasoner)** complete tasks that **You
     (reasoner)** cannot perform directly, including:
{task_category}
Our goal is to resolve the **User (requester)**'s request step by step
    .
* If **You (reasoner)** want to assign a task to **Me (executor)**,
    please use the <task>...</task> tag to describe your task, don't
    give the detail execution commands for now.
* If **You (reasoner)** want to request information from the **User (
    requester)**, ask them directly without using any tags.
If you believe the user's request has already been resolved, please
    answer the **User (requester)**'s request based on the entire
    conversation history and at the end of your answer add <finish>
    exit</finish>.
Here is an example:
USER: <user_request>
Can you help me to download a PDF file called example.pdf from the
    internet and convert its Table 1 to .csv file?
</user_request>
ASSISTANT: <task>Download the example.pdf from the internet.</task>
USER: [Downloads output]
ASSISTANT: <task>Convert Table 1 of example.pdf to .csv file.</task>
USER: [Converts output]
ASSISTANT: <task>Verify if the .csv file exist.</task>
USER: [Terminal output]
ASSISTANT: Now the task is finished and we can provide the .csv file
    to the user. <finish>exit</finish>
```

Listing 2: Planning end prompt. It is appended to the end of each planning.

```
If you believe the user's request has already been resolved, please
    answer the **User (requester)**'s request based on the entire
    conversation history and at the end of your answer add <finish>
    exit</finish>.
Otherwise, provide assign a task to **Me (executor)** within the
    appropriate execution tag:
> Use the <task>...</task> tag for the task that you would like to
    assign to me.
Please do NOT repeat the similar analysis that you have already
    provided.
```

Listing 3: Planning avoid repetition prompt. It is appended to the end of the planning dialogue whenever the planning process is needlessly repeated.

```
If you believe the user's request has already been resolved, please
    answer the **User (requester)**'s request based on the entire
    conversation history and at the end of your answer add <finish>
    exit</finish>.
Otherwise, please assign a task based on your analysis.
Please do NOT repeat the similar analysis/Task again!
```

## B.2 Tool Selection

Listing 4: Tool Selection System Prompt. it is inserted at the beginning of each tool selection action.

```
I would like to finish a task, please help me choose the suitable sets
    of commands to complete this task.
1. File editing related commands:
This set of commands can be used to view file content, as well as
    perform additions, deletions, searches, and modifications on files
    .
If you want to select this set of commands, please return: <toolkit>
    file_edit</toolkit>
2. Code execution related commands:
This set of commands can be used to execute code snippets.
If you want to select this set of commands, please return: <toolkit>
    code_exec</toolkit>
3. Computer interaction commands:
These commands can be used to interact with the computer via the
    keyboard and mouse.
If you want to select this set of commands, please return: <toolkit>
    computer_interaction</toolkit>
4. Web browsing related commands:
This set of commands can be used to interact with web pages.
If you want to select this set of commands, please return: <toolkit>
    web_browse</toolkit>
5. File understanding related commands:
This set of commands can be used to understand the content of files.
    Such as reading files, view images, listen to audios, watch videos
    , etc.
If you want to select this set of commands, please return: <toolkit>
    file_understand</toolkit>
If you want to select multiple sets of commands, please separate them
    with commas.
For example, if you think we not only need to edit some files but also
     execute some code, you should return: <toolkit>file_edit,
    code_exec</toolkit>.
```

## B.3 Execution

Listing 5: Execution system prompt. it is inserted at the beginning of execution. It includes usage instructions and examples for the selected tool functions.

```
{Selected_Tool_Guide_Book}
{Selected_Tool_Examples}
```

Listing 6: Execution end prompt. it is appended to the end of the execution.

```
If you think the current task: {task} is already solved, please
    respond with your conclusion and include the following tag at the
    end:
<task_finish>
exit
</task_finish>.
Otherwise, provide the next command within the appropriate execution
    tag:
> Use <execute_bash>...</execute_bash> for Bash commands.
> Use <execute_python>...</execute_python> for my other customized
    commands, as I mentioned in the beginning.
Please don't take any steps on your own that aren NOT related to
    completing the current task: {task}; I will guide you through the
    next steps.
You only need to ensure that the current task is completed.
```

# C  Toolkits

## C.1  File Edit Toolkit

Listing 7: File edit tool functions

```
Please use the following file editing functions to add, delete, search
    , and modify files.
- create_file(filename: str, content: str | None): Creates and opens a
     new file with the given name. Add the content to the file if
    content is not None.
- replace_content(file_path, old_content, new_content): Replaces the
    old content with the new content in the specified file. For the
    old_content/new_content argument, please focus only on the parts
    that actually need to be changed.
- edit_file(file_name: str, start_line: int, start_str: str, end_line:
     int, end_str: str, content: str): Edits the specified file by
    replacing the content between start and end lines with the new
    content. file_name: Name of the file. start_line: Starting line
    number. start_str: String content in Starting line. end_line:
    Ending line number. end_str: String content in Ending line.
    content: New content to replace.
- append_file(file_name, content, start_line): Appends given content
    to a file. file_name: Name of the file. content: Content to append
    . start_line: Line number to start appending from (default is the
    end of the file).
- search_function(file_path, function_signature): Search and show a
    function in the file. For the function_signature, you should only
    input the function name.
```

## C.2  File View Toolkit

Listing 8: File view tool functions

```
Please use the following functions to understand the content of files.
     Such as reading files, view images, listen to audios, watch
    videos, etc.
- open_file(path: str, line_number: int | None = 1, context_lines: int
     = 100): Opens a file (txt, csv, word, code file, etc.) and
    optionally moves to a specific line. path: Path to the file.
    line_number: Line number to move to. context_lines: Number of
    lines to display.
- parse_pdf(pdf_path: str, page: int): View the specified page of a
    PDF file. pdf_path: Path to the PDF file. page: Page number to
    view.
- parse_figure(figure_path: str): View the specified figure.
    figure_path: Path to the figure file.
- parse_audio(audio_path: str, question: str): Ask a question about
    the audio file. audio_path: Path to the audio file. question:
    Question to ask.
- zoom_pdf(pdf_path: str, page: int, region: tuple): Zoom in on a
    specific region of a PDF file. pdf_path: Path to the PDF file.
    page: Page number to view. region: Tuple specifying the region to
    zoom in on (x0, y0, x1, y1).
```

## C.3  File Search Toolkit

Listing 9: File search tool functions

```
Please use the following functions to search files.
- search_dir(search_term, dir_path='./'): Searches for a term in all
    files in the specified directory.
```

```
- find_file(file_name, dir_path='./'): Finds all files with the given
    name in the specified directory.
```

## C.4   Web Browser Toolkit

Listing 10: Web browser tool functions

```
You can use the following functions to interact with the browser.
- open_browser(): Open the browser.
- navigate_to(url: str) : Navigate to the specified URL.
- refresh_page(): Refresh the current page.
- go_back(): Go back to the previous page.
- go_forward(): Go forward to the next page.
- close_current_tab(): Close the current tab.
- execute_javascript(script: str): Execute the specified JavaScript
    code.
- switch_to_tab(page_id: int): Switch to the tab at the specified
    index.
- create_new_tab(url: str): Open a new tab with the specified URL.
- save_cookies(): Save the current cookies.
- select_dropdown_option(selector_index: int, option: int): Select the
    specified option from the dropdown menu. selector_index: selector
    index. option: Index of the option to select.
- google_search(content: str): Perform a Google search for the
    specified content. content: The content to search for.
- close(): Close the browser.
```

## C.5   Computer Use Toolkit

Listing 11: Computer use functions

```
You can use the following functions to perform various mouse and
    keyboard operations.
- clear_text(): Clear the text in the current input field. please make
    sure the input field is selected before using this command.
- take_screenshot(): If you want to check the current screen, you can
    use this command to take a screenshot of the current screen.
- mouse_left_click(item: str, description: str): Left mouse click at
    the specified position. For example: mouse_left_click('search bar
    ', 'It is located near the top center of the Google Chrome browser
     window. It is a long, rectangular input field with rounded edges.
     The search bar spans almost the entire width of the browser
    window and sits directly below the browser's tab row. It has
    placeholder text that reads "Search Google or type a URL." The
    search bar is centrally aligned, making it easy to spot above the
    main content area of the browser.')
- mouse_double_click(item: str, description: str): Double-click at the
    specified position. For example: mouse_double_click('The VSCode
    icon', 'It is located in the sidebar (Launcher) on the left side
    of the screen. It is the first icon from the top in the vertical
    arrangement. The icon has a blue background with a white folded "V
    "-shaped design in the center. The sidebar is aligned along the
    leftmost edge of the screen, adjacent to the desktop background on
     its right side.')
- mouse_right_click(item: str, description: str): Right mouse click at
    the specified position. For example: mouse_right_click('The
    refresh button', 'It is located at the top-left corner of the
    Google Chrome browser window, inside the address bar. It is a
    circular arrow icon situated next to the left and right navigation
     arrows (back and forward buttons). The refresh button is just to
    the left of the search bar. Click it to refresh the current page
    .')
```

```
- mouse_scroll(direction: str, amount: int): Scroll mouse scroll up or
     down. direction: Direction to scroll ("up" or "down"). amount:
   Number of times to scroll.
- type_text(text: str): Type the given text. text: The text to type.
- download(url: str, save_dir: str): If you know the url of the file,
   you can use this command to download the file from the specified
   URL to the specified directory.
- press_key(key: str): Presses the specified key. key: The key or key
   combination to press (e.g., "Return", "Ctrl+c").
- open_application(app_name: str): Opens a specific application using
   the system application launcher. app_name: The name of the
   application to open (e.g., "chrome").
- mouse_drag(x_start: int, y_start: int, x_end: int, y_end: int): Drag
    the mouse from one position to another. x_start: Starting x-
   coordinate. y_start: Starting y-coordinate. x_end: Ending x-
   coordinate. y_end: Ending y-coordinate.
- mouse_box_select(x_start: int, y_start: int, x_end: int, y_end: int)
   : Selects a box by dragging the mouse from one position to another
   . x_start: Starting x-coordinate. y_start: Starting y-coordinate.
   x_end: Ending x-coordinate. y_end: Ending y-coordinate.
```

## C.6 Code Execution Toolkit

Listing 12: Code execution tool

```
If no suitable command is available, you can also use bash commands to
    interact with the terminal.
Do not use bash commands to edit/create the files. Instead, use the
   file editing functions provided.
```

## C.7 Advanced Toolkit

Listing 13: Advanced tools

```
You can use the following functions to perform advanced operations.
   These commands are compound commands. When available, please
   prefer using these commands rather than attempting to perform the
   operations by yourself.
- search_arxiv(keyword: str, start_date: str, end_date: str, subject:
   str, field: str) : Searches for papers on arXiv based on the given
    keyword, date range, subject (options: cs, math, physics, q-bio,
   q-fin, stat), and keyword field (options: title, abstract,
   comments, author, all). Returns the search results.
- download_arxiv_pdf(arxiv_id: str) : Downloads the specified arXiv
   paper based on its ID (eg: 1608.06816) and show the first page of
   the PDF.
- scroll_pdf_page(direction: str, pages: int): When you're viewing a
   PDF document in the web interface, you can use this command to
   scroll the specified page of the PDF file up or down. direction:
   Direction to scroll ("up" or "down"). page: Page number to scroll.
- watch_video(video_path_or_url: str): Please use this command to
   Watch a video file or YouTube URL. Especially when there is a sign
   -in prompt for Youtube. video_path_or_url: Local path or YouTube
   URL.
- count_string_in_pdf(pdf_path: str, search_string: str): Count the
   number of occurrences of a specific string in a **Local** PDF file
   . pdf_path: Path to the PDF file. search_string: The string to
   search for.
.
```

# D Details of Dedicated Computer

A fully isolated execution environment enhances security and enables full-spectrum command execution. Many existing agents (e.g., AutoGPT [18]) run directly on the host machine or run CLI operations within Docker containers. In contrast, our INFANTAGENT-NEXT provides a dedicated, standalone computing environment that supports command-line interaction via Bash, Python scripts via Jupyter, and direct GUI operations through the GNOME desktop. Specifically, to ensure safety and control, the agent operates within an isolated virtual machine (VM), preventing direct access to the user's computer and thus eliminating the risk of modifying or deleting critical files. At the same time, the user can still interact with the VM. The setup consists of three stages: (1) Build a base Docker image; (2) Enable GPU-accelerated rendering to support high frame rates and image quality; (3) Expose the VM's GNOME desktop via a remote desktop interface. When the agent requires visual feedback (e.g., to perform a mouse click), the VM captures a screenshot of the current display and sends it back to the agent. The agent then analyzes the image before issuing the next command.

# E Test Cases of SWE-Bench-Verified

We detail the test cases of SWE-Bench-Verified in the following:

- `astropy__astropy-12907`
- `astropy__astropy-14995`
- `astropy__astropy-7606`
- `astropy__astropy-8707`
- `django__django-11451`
- `django__django-11603`
- `django__django-12858`
- `django__django-13417`
- `django__django-14500`
- `django__django-15930`
- `django__django-16032`
- `django__django-16256`
- `django__django-16899`
- `matplotlib__matplotlib-20859`
- `matplotlib__matplotlib-22719`
- `matplotlib__matplotlib-24970`
- `matplotlib__matplotlib-25122`
- `mwaskom__seaborn-3069`
- `mwaskom__seaborn-3187`
- `pallets__flask-5014`
- `psf__requests-1142`
- `psf__requests-1766`
- `psf__requests-1921`
- `psf__requests-5414`
- `pydata__xarray-4075`
- `pydata__xarray-6599`
- `pydata__xarray-6744`
- `pydata__xarray-6938`
- `pylint-dev__pylint-4661`

- `pylint-dev__pylint-4970`
- `pylint-dev__pylint-6386`
- `pylint-dev__pylint-6528`
- `pytest-dev__pytest-10081`
- `pytest-dev__pytest-5631`
- `pytest-dev__pytest-5809`
- `pytest-dev__pytest-7205`
- `pytest-dev__pytest-7432`
- `scikit-learn__scikit-learn-10297`
- `scikit-learn__scikit-learn-12585`
- `scikit-learn__scikit-learn-12973`
- `scikit-learn__scikit-learn-13135`
- `sphinx-doc__sphinx-11445`
- `sphinx-doc__sphinx-7454`
- `sphinx-doc__sphinx-8035`
- `sphinx-doc__sphinx-8551`
- `sphinx-doc__sphinx-8721`
- `sympy__sympy-13798`
- `sympy__sympy-14531`
- `sympy__sympy-16792`
- `sympy__sympy-17630`

## F   Limitations and Further Work

Our current work is confined to the reasoning phase. To mitigate the impact of excessive prompt engineering, we will next further train the model to automatically invoke the appropriate tools rather than relying on manually added prompts.

