# OpenReview forum: "InfantAgent-Next: A Multimodal Generalist Agent for Automated Computer Interaction"
_NeurIPS.cc/2025/Conference — NeurIPS 2025 poster_

### Official Review · Reviewer_x34C · 2025-06-30

**Clarity:** 4
**Significance:** 3
**Originality:** 2
**Rating:** 5
**Confidence:** 5

**Summary:**

The paper introduces INFANTAGENT-NEXT, a multimodal generalist agent designed for automated computer interaction, capable of handling text, images, audio, and video inputs. It proposes a modular architecture that integrates tool-based and vision-based approaches, routing subtasks to specialized models (e.g., reasoning, visual grounding, audio analysis) within a unified dialogue context. The agent achieves strong performance across diverse benchmarks, including OSWorld (7.27% accuracy, surpassing Claude-Computer-Use), SWE-Bench (66% on a verified subset), and GAIA (second among open-source agents on Level 2 tasks). The work includes open-source code and evaluation scripts, emphasizing reproducibility and modularity for tasks like file editing, web browsing, and GUI interaction.

**Questions:**

1. **Failure Case Analysis**: The paper mentions minor limitations in file editing edge cases (e.g., 9.6% failure rate). Could the authors provide a detailed analysis of these failures, including specific examples and potential improvements? This could strengthen the quality score by demonstrating robustness.
2. **Scalability and Cost**: The paper uses resource-intensive models (e.g., Claude-3.7-Sonnet, A100 GPUs). How does INFANTAGENT-NEXT scale for real-world deployment, particularly in terms of computational cost and latency? Addressing this could elevate the significance score.
3. **Novelty of Workflow**: The modular workflow is effective but builds on existing paradigms. Can the authors clarify what aspects of the workflow (beyond integration) are truly novel? A stronger case for originality could improve the originality score.

**Ethical Concerns:**

["NO or VERY MINOR ethics concerns only"]

**Limitations:**

Yes

**Paper Formatting Concerns:**

**Issues**: None
**Justification**: The paper adheres to NeurIPS formatting guidelines, with proper structure, citations, and supplemental material.

**Quality:**

4

**Strengths And Weaknesses:**

**Strengths:**
- **Quality**: The paper presents a robust engineering effort with a well-designed modular workflow that effectively combines tool-based and vision-based paradigms. Extensive evaluations across OSWorld, SWE-Bench, and GAIA demonstrate strong empirical performance, with INFANTAGENT-NEXT outperforming notable baselines like Claude-Computer-Use and achieving competitive results among open-source agents. The inclusion of detailed algorithms (e.g., Iterative Region Cropping, File Editing Logic) and open-source code enhances reproducibility.
- **Clarity**: The paper is clearly written, with a structured breakdown of the architecture, components, and experimental setups. Figures and case studies (e.g., Figure 2 and 5) effectively illustrate the workflow, and the appendix provides comprehensive details on prompts, toolkits, and test cases.
- **Significance**: The work is highly significant for the field of automated AI agents, addressing practical challenges in multimodal computer interaction. Its broad applicability across GUI tasks, coding, and general tool-use benchmarks makes it a valuable contribution to real-world agent development.

**Weaknesses:**
- **Originality**: While the engineering is impressive, the paper lacks novel theoretical insights or fundamentally new paradigms in agent design. The modular architecture, while effective, builds on existing concepts (e.g., tool-based and vision-based agents) without introducing groundbreaking ideas.
- **Quality (Limitation)**: The evaluation, though comprehensive, could benefit from deeper analysis of failure cases or edge scenarios to better understand the agent’s limitations. For instance, the paper notes minor limitations in file editing edge cases but does not explore these systematically.
- **Significance (Scope)**: The work focuses heavily on performance metrics but provides limited discussion on scalability or real-world deployment challenges, such as computational costs for large-scale use or robustness in dynamic environments.

---

> ### Author Rebuttal · Authors · 2025-07-31
>
> Thank you for your positive assessment of our work! Below, we provide point‑by‑point responses to the comments.
> ***********
> **For weakness 1 and Question 3:**
>
> > the paper lacks novel theoretical insights or fundamentally new paradigms in agent design.
> When integrating pure-vision agents with tool‑based agents to build multimodal, general‑purpose agents, we encounter several categories of challenges not present in prior single‑paradigm agents. Below, we describe each challenge in turn and present our novel solutions.
> ***********
> ***Challenge 1*** — Growth in tool count inflates inference cost.
>
> Traditional tools‑based and pure‑vision agents each expose their own action space. For example, OpenHands (tool‑based) offers file‑editing commands such as `file_edit` and `file_open`, whereas UI‑TARS (pure‑vision) provides mouse/keyboard primitives like `mouse_click` and `press_key`. When these paradigms are fused, we must unify all action spaces and, to handle additional modalities (e.g., audio/video), introduce extra tools. All of these tools—together with usage examples—need to be registered at the start of the dialogue (e.g., in the tool manifest/system prompt), which substantially increases context length and token usage, thereby raising inference cost and latency relative to single‑paradigm agents.
>
> ***Our novel method for challenge 1***:
> We introduce a tool‑selection step  (**Fig. 2** and Lines 109–118): after the planner produces a subtask plan, the agent performs task‑conditioned tool selection, activating only the minimal set of tools needed for the upcoming subtask.
>
> Original workflow:
> ```
> agent.register_tools()        # registers/activates everything up front
> await agent.planning()
> await agent.execution()
> ```
> Our workflow:
> ```
> await self.planning()
> await self.tool_selection() # selects a minimal tool set for this subtask
> await self.execution()
> ```
>
> ***Example***
> As illustrated in **Fig. 5**, the three tool‑selection phases (blue shaded regions) choose different categories of tools conditioned on the current subtask, rather than registering the entire tool inventory to the model. This selective exposure reduces inference cost.
> ***********
> ***Challenge 2*** — Rising task complexity vs. limited model capacity.
>
> Traditional tool‑based agents typically rely on a single backbone model to handle an entire task—AutoGPT and BabyAGI, for instance, route planning and execution through one (or at most a two‑slot fast/slow) LLM—whereas some vision‑based agents augment the main model with a dedicated GUI visual‑grounding component for pointer‑level localization, as in OS‑Atlas. However, real‑world multimodal agents must operate across text, images, audio, and video. Given this breadth and cross‑domain complexity, a single model—or a small fixed set of models—is often insufficient to deliver accurate and reliable performance.
>
> ***Our novel method for challenge 2***:
> We leverage the agent’s modular design: the end-to-end workflow is decomposed into well-defined modules, and subtasks are routed to specialized models. Difficult, cross-domain steps are assigned to large, high-capability backbones—e.g., we rely on Claude-3.7-Sonnet for global task planning. We assign latency-sensitive operations to lower-latency models; e.g., Qwen2.5-72B-Instruct is used for precise file-editing commands. Vision subtasks go to vision models, and audio subtasks to audio models. This task‑conditioned model routing preserves accuracy while reducing cost and latency. Implementation details are provided in **Fig. 2** and Lines 101–108.
>
> ***Example***. In **Fig. 5**, the execution step beneath Turn 1 (purple region) corresponds to image analysis performed by a vision model, whereas the execution steps under Turns 2–3 are carried out by a code‑specialized model.
> This setup exploits complementary strengths and reduces failures due to modality limits or capability shortfalls.
>
> ***********
> ***Challenge 3*** — Tool‑use hallucination
>
> Even SOTA models can hallucinate when invoking tools that require precise numeric arguments. Typical failure modes include providing an incorrect line number to the file‑editing tool or wrong coordinates to the visual‑grounding tool. For example, in our tests reported in **Sec. 4.5**, DeepSeek‑V3‑0324 shows a $61\\%$ tool‑use hallucination rate on file‑editing commands without any mitigation; in long‑horizon multimodal tasks, such small errors compound and can ultimately lead to end‑to‑end failure.
>
> ***Our novel method for challenge 3***: We design two algorithms tailored to these error modes that substantially reduce tool‑use hallucinations.
>
> **Alg. 1** (Iterative Region Cropping & Mouse Click): Repeatedly ground → crop → re-ground to progressively narrow the region before the final click, reducing misclicks without retraining and working with any GUI grounder.
>
> **Alg. 2** (File Editing: Plan-Verify-Repair): Have the model propose edit boundaries with content anchors, verify them against the file, and on mismatched fuzzy re-localize and repair before applying the edit, curbing line-number hallucinations.
>
> see **Alg. 1** and **Alg. 2** for details.
>
> ***Effectiveness of method 3***. In our ablation studies (**Sec. 4.4** and **Sec. 4.5**), **Alg. 1** increases visual‑grounding accuracy on high‑resolution images by $11.6$ percentage points, while **Alg. 2** corrects $84.3\\%$ of file‑editing tool sequencing errors.
>
> **For weakness 2 and Question 1:**
>
> > The paper mentions minor limitations in file editing edge cases (e.g., 9.6% failure rate). Could the authors provide a detailed analysis of these failures, including specific examples and potential improvements? This could strengthen the quality score by demonstrating robustness.
>
> To mitigate hallucinations that arise when the model uses the file‑editing tool, we introduce **Alg. 2** and analyze its effectiveness in correcting different error types in **Sec. 4.5**; the results are visualized in **Fig. 4**.
> Here we will show a specific example:
>
> Task: Edit on `astropy/wcs/wcs.py`, Agent is trying to add a debug statement between line 1249 and line 1250.
>
> ```
>  ... (1242 more lines above)
> 1246 | if len(args) == 2:
> 1247 |     try:
> 1248 |         xy, origin = args
> 1249 |         xy = np.asarray(xy)
> 1250 |         origin = int(origin)
> 1251 |         if xy.size == 0:
> 1252 |             return np.array([])
> ... (2040 more lines below)
> ```
>
> ********** **Original Incorrect Action** ********** ❌
> ```
> edit_file(astropy/wcs/wcs.py,
> start_line = 1249,
> end_line = 1249,   # ⚠️ This line number is wrong, should be 1250.
> content = "         print(f'xy:{xy}')")
> ```
> ********** **Our method** ********** ✔️
>
> Step 1: The model is asked to produce the content corresponding to the line it plans to edit.
> ```
> edit_file(astropy/wcs/wcs.py,
> start_line = 1249,
> start_line_str = 'xy = np.asarray(xy)'
> end_line = 1249,   # ⚠️ if this line number is not correct
> end_line_str = 'origin = int(origin)' # ⚠️ the content will not match
> content = "         print(f'xy:{xy}')")
> ```
>
> Step 2: Upon detecting a mismatch between the model‑generated content and the actual content at line 1249, we display a diagnostic message:
>
> `The string: origin = int(origin) does not match the end line 1249 The end line: 1249 is: xy = np.asarray(xy)`
>
> Step 3: The model regenerates the command:
> ```
> edit_file(astropy/wcs/wcs.py,
> start_line = 1249,
> start_line_str = 'xy = np.asarray(xy)'
> end_line = 1250,  # Successfully corrected the line number. ✔️
> end_line_str = 'origin = int(origin)'
> content = "         print(f'xy:{xy}')")
> ```
>
> **For weakness 3 and Question 2:**
> > How does INFANTAGENT-NEXT scale for real-world deployment, particularly in terms of computational cost and latency? Addressing this could elevate the significance score.
>
> We are actively deploying the system in real settings. Due to space constraints, we summarize two practical tactics we use to reduce computational cost at scale:
>
> 1. **Preserve the KV cache.** Keep the prompt/tool schema as stable as possible so cached attention states remain reusable across turns. On hosted APIs (e.g., OpenAI/Claude), “cached tokens” are typically billed at ≈$25$–$50\\%$ of the price of new tokens; on self‑hosted models, retaining the KV/attention cache also yields substantial latency speedups. **Concretely, we mask tools that are not needed in a given step rather than deleting them** (and we keep tool order and identifiers fixed), which maintains cache compatibility from one turn to the next.
>
> 2. **Offload verbose tool outputs to files (workspace artifacts)**. Tool calls often return long texts. Instead of appending the entire output to the chat history, we write it to a file in the agent’s workspace and refer to it by a short handle (path/hash) and a brief summary. The agent re‑opens the file on demand when needed. This keeps the conversation context short, reducing both token cost and inference latency.
>
> ***********
> We once again thank the reviewer for the thoughtful comments. We hope our responses address your concerns, and we would be glad to continue the discussion should any further questions arise.

---

> > ### Comment · Reviewer_x34C · 2025-08-05
> >
> > Thank you for the detailed rebuttal. However, I maintain my assessment regarding novelty.
> >
> > **Challenges 1 & 2** (tool selection and model routing) are essentially prompt-based agent context management problems that have been extensively explored since 2022 and are now integrated into mature industrial frameworks like Claude agents, Claude Code, and Cursor.
> >
> > **Challenge 3** (regrounding): The iterative region cropping approach has been seen in GUI-related work like ScreenSpot-Pro (2025.1), and similar methods likely appeared earlier in multimodal/CV literature.
> >
> > I continue to view this as an effective combination of existing techniques achieving competitive benchmark results with strong engineering value - which I do recognize and appreciate, as reflected in my high Quality and Clarity scores.
> >
> > Given that my initial rating already acknowledges the work's merits (Accept), I will not modify my scores.
> >
> > Thank you for your contributions and open-source release.

---

> > > ### Author Response · Authors · 2025-08-05
> > >
> > > Thank you for your positive feedback on our work! We will continue to improve our agent based on your valuable suggestions and hope to make meaningful contributions to the development of the relevant community.

---

### Official Review · Reviewer_LrsF · 2025-07-02

**Clarity:** 3
**Significance:** 2
**Originality:** 2
**Rating:** 4
**Confidence:** 4

**Summary:**

This paper introduced a generalist agent framework named InfantAgent-Next. The framework adopts a multi-model collaboration architecture, where models with different modalities and specializations are assigned to various tasks (such as planning, tool selection, and visual grounding) to achieve optimization of overall capabilities. Experiments demonstrate that InfantAgent-Next achieves promising results across multiple types of tasks, including computer interaction (OSWorld), real-world question answering (GAIA), and coding (SWE-Bench).

**Questions:**

1. Could the author elaborate more on the methodological innovation and contributions? While engineering optimization is crucial for building such an agent system, as an academic paper, we would value an analysis of the underlying principles and methodological advancements.

2. In Table 1, the author compares the proposed method with existing works from the perspective of modular design. Could the author provide an ablation analysis of all these modules in InfantAgent-Next?

3. InfantAgent-Next is designed to be model-agnostic and flexible for use with various LLMs/vLLMs. The experiments only tested Claude-3.7-Sonnet and UI-TARS-1.5-7B—how significantly would performance change with different models?

**Ethical Concerns:**

["NO or VERY MINOR ethics concerns only"]

**Final Justification:**

This paper introduced InfantAgent-Next, a generalist computer-use agent that achieves strong performance on multiple benchmarks. Although this work lacks remarkable innovation in methodology, it offers engineering insights for building efficient computer-use agents by integrating multimodal models and technologies like tool selection and memory retrieval. I believe this work aligns with the infrastructure category of the call-for-papers and will benifit future research.

**Limitations:**

yes

**Quality:**

3

**Strengths And Weaknesses:**

# Strengths

1. The proposed agent system is evaluated on multiple benchmarks and achieves promising results on tasks such as computer interaction and real-world problem-solving. This demonstrates the versatility of the proposed method. And I believe it can be readily extended to a broader range of scenarios and tasks, considering its flexibility in integrating diverse models and tools.

# Weaknesses

1. This work is more of an integration of existing technologies. The design of the agent system's architecture and workflows does not exhibit significant innovations compared to existing methods; the design of core modules, such as planning, visual grounding, tool use, and context management, is also not original.

2. I also have concerns about the fairness of the experiment results. Considering that the contribution of this work lies in the agent architecture rather than the model itself, the comparison should focus on the agent design. However, the baseline methods in table 2-4 are based on different models, some of which have significant gaps in parameter size and model capability.

---

> ### Author Rebuttal · Authors · 2025-07-31
>
> We thank the reviewer for the thoughtful and constructive feedback. Below, we provide point‑by‑point responses to the comments.
> ***********
> **For weakness 1 and Question 1:**
> > Could the author elaborate more on the methodological innovation and contributions?
>
> **1. Our contributions:**
>
> Based on the NeurIPS 2025 Call for Papers taxonomy, we position InfantAgent-Next in the **Infrastructure category** (e.g., libraries, improvements in implementation and scalability, and distributed solutions). We have two main contributions:
>
> - Infrastructure for computer-use agents. **We present an infrastructure for building multimodal, general-purpose agents with integrated computer-use capability**—i.e., agents that can operate their own computers.
>
>
> - Implementation improvements via unification and interfaces. **We unify tool-based and purely vision-based approaches into a single stack and expose unified interfaces that broaden agents’ computer-use capabilities**. We hope this serves as a versatile reference for the community.
>
> **2. Our novel methods for new challenges**
>
> When integrating pure-vision agents with tool‑based agents to build multimodal, general‑purpose agents, we encounter several categories of challenges not present in prior single‑paradigm agents. Below, we describe each challenge in turn and present our novel solutions.
> ***********
> ***Challenge 1*** — Growth in tool count inflates inference cost.
>
> Traditional tools‑based and pure‑vision agents each expose their own action space. For example, OpenHands (tool‑based) offers file‑editing commands such as `file_edit` and `file_open`, whereas UI‑TARS (pure‑vision) provides mouse/keyboard primitives like `mouse_click`. When these paradigms are fused, we must unify all action spaces and, to handle additional modalities (e.g., audio/video), introduce extra tools. All of these tools—together with usage examples—need to be registered at the start of the dialogue (e.g., in the tool manifest/system prompt), which substantially increases context length and token usage, thereby raising inference cost and latency relative to single‑paradigm agents.
>
> ***Our novel method for challenge 1***:
> We introduce a tool‑selection step  (**Fig. 2** and Lines 109–118): after the planner produces a subtask plan, the agent performs task‑conditioned tool selection, activating only the minimal set of tools needed for the upcoming subtask.
>
> Original workflow:
> ```
> agent.register_tools()        # registers/activates everything up front
> await agent.planning()
> await agent.execution()
> ```
> Our workflow:
> ```
> await self.planning()
> await self.tool_selection() # selects a minimal tool set for this subtask
> await self.execution()
> ```
>
> ***Example***
> As illustrated in **Fig. 5**, the three tool‑selection phases (blue shaded regions) choose different categories of tools conditioned on the current subtask, rather than registering the entire tool inventory to the model. This selective exposure reduces inference cost.
> ***********
> ***Challenge 2*** — Rising task complexity vs. limited model capacity.
>
> Most tool-based agents run end-to-end on a single LLM (e.g., AutoGPT, BabyAGI), and some vision-based agents add a GUI visual-grounding module for pointer actions (e.g., OS-Atlas). However, real-world multimodal tasks span text, images, audio, and video, so a single—or small fixed set of—models rarely delivers accurate, reliable performance.
>
> ***Our novel method for challenge 2***:
> We leverage the agent’s modular design: the end-to-end workflow is decomposed into well-defined modules, and subtasks are routed to specialized models. Difficult, cross-domain steps are assigned to large, high-capability backbones. We assign latency-sensitive operations to lower-latency models. Vision subtasks go to vision models, and audio subtasks to audio models. This task‑conditioned model routing preserves accuracy while reducing cost and latency. Implementation details are provided in **Fig. 2** and Lines 101–108.
>
> ***Example***. In **Fig. 5**, the execution step beneath Turn 1 (purple region) corresponds to image analysis performed by a vision model, whereas the execution steps under Turns 2–3 are carried out by a code‑specialized model.
> This setup exploits complementary strengths and reduces failures due to modality limits or capability shortfalls.
>
> ***********
> ***Challenge 3*** — Tool‑use hallucination
>
> Even SOTA models can hallucinate when invoking tools that require precise numeric arguments. Typical failure modes include providing an incorrect line number to the file‑editing tool or wrong coordinates to the visual‑grounding tool. For example, in our tests reported in **Sec. 4.5**, DeepSeek‑V3‑0324 shows a $61\\%$ tool‑use hallucination rate on file‑editing commands without any mitigation; in long‑horizon multimodal tasks, such small errors compound and can ultimately lead to end‑to‑end failure.
>
> ***Our novel method for challenge 3***: We design two algorithms tailored to these error modes that substantially reduce tool‑use hallucinations.
>
> **Alg. 1** (Iterative Region Cropping & Mouse Click): Repeatedly ground → crop → re-ground to progressively narrow the region before the final click, reducing misclicks without retraining and working with any GUI grounder.
>
> **Alg. 2** (File Editing: Plan-Verify-Repair): Have the model propose edit boundaries with content anchors, verify them against the file, and on mismatched fuzzy re-localize and repair before applying the edit, curbing line-number hallucinations.
>
> see **Alg. 1** and **Alg. 2** for details.
>
> ***Effectiveness of method 3***. In our ablation studies (**Sec. 4.4** and **Sec. 4.5**), **Alg. 1** increases visual‑grounding accuracy on high‑resolution images by $11.6$ percentage points, while **Alg. 2** corrects $84.3\\%$ of file‑editing tool sequencing errors.
>
> **For weakness 2:**
> > ….some of which have significant gaps in parameter size and model capability.
>
> In **Table 2–4**, we present comparisons between InfantAgent‑Next and existing agents, with the aim of demonstrating competitive performance across a range of tasks.
>
> For several baselines, using a single, unified backbone is infeasible. For example, in **Table 2**, UI‑TARS reports results with its GUI‑specialized model (UI‑TARS‑1.5‑72B), rather than a public backbone. In **Table 4**, OWL processes webpages by converting them to HTML, whereas our framework operates directly on screenshots. Consequently, a text‑only backbone without visual capability is incompatible with our setting and would substantially degrade InfantAgent‑Next’s performance.
>
> For baselines where a unified backbone is feasible, we provide comparisons under the same model. For example, in **Table 2** we match AgentS2 by using the same model and enforcing the same step budget; in **Table 4**, we likewise compare with TapeAgents/Auto‑Deep‑Research under the same model. This controls for backbone effects and isolates the contribution of our framework.
>
> **For Question 2**
>
> > ... Could the author provide an ablation analysis of all these modules in InfantAgent-Next?
>
> **Table 1** contrasts InfantAgent-Next with prior agents across eight dimensions. We do not ablate four of them—Generalist Agent, Multi-Model Support, User Interaction during Execution, and Dedicated Computer—because they are problem-setting assumptions, not pluggable modules; toggling them would alter the evaluation protocol.
>
> Accordingly, we ablate the remaining modules by disabling each in turn within InfantAgent-Next and evaluating 50 GAIA cases, reporting success rate and per-task cost; Claude-3.7-Sonnet serves as the backbone.
>
> | Ablation             | Success rate (%) | Average cost per task ($) |Analysis of Results   |
> |:--------------------:|:----------------:|:--------------------------:|:-----------------------:|
> | Full                 | $72$               | $2.77$                       | baseline                |
> | Visual Grounding off | $44$               | $2.43$                       | pointer tasks drop      |
> | Tool off             | $12$               | $2.15$                       | File operation fail     |
> | Memory Retrieval off | $78$               | $3.89$                       | Inference cost increase |
> | Dynamic Toolset off  | $72$               | $4.54$                       | Inference cost increase |
>
> **For Question 3:**
> > ... how significantly would performance change with different models?
>
> We evaluated three backbone models: Claude‑3.7‑Sonnet, GPT‑4o, and DeepSeek‑V3‑0324. Because DeepSeek‑V3‑0324 lacks multimodal ability, we used it exclusively on the SWE‑Bench test. Since agent performance is strongly coupled with the backbone model’s capability, in our main comparisons against SOTA agents we report results using the strongest backbone per setting.
>
> Below we provide our measured results for GPT‑4o and DeepSeek‑V3‑0324:
>
> **SWE‑bench‑Verified test results.**
> | Agent            | Model            | Accuracy |
> |:----------------:|:----------------:|:-------:|
> | OpenHands        | DeepSeek-V3-0324 |    $38.8\\%$ |
> | Agentless        | DeepSeek-V3-0324 |    $35.8\\%$ |
> | AutoCodeRover    | DeepSeek-V3-0324 |    $34.8\\%$ |
> | **InfantAgent-Next** | DeepSeek-V3-0324 |    $37.2\\%$ |
> | OpenHands        | GPT-4o           |   $41.20\\%$ |
> | Agentless        | GPT-4o           |   $38.80\\%$ |
> | AutoCodeRover    | GPT-4o           |   $39.20\\%$ |
> | **InfantAgent-Next** | GPT-4o           |   $40.20\\%$ |
>
> **OSWorld test results using GPT‑4o as the backbone model (maximum 50 steps per task).**
>
> | Agent            | Accuracy |
> |:----------------:|:-------:|
> | Agent S          |   $20.58\\%$ |
> | OpenAI CUA       |    $32.6\\%$ |
> | Agent S2         |   $27.91\\%$ |
> | Aguvis           |   $17.04\\%$ |
> | **InfantAgent-Next** |   $30.35\\%$ |
>
> ***********
> We once again thank the reviewer for the thoughtful comments. We hope our responses address your concerns, and we would be glad to continue the discussion should any further questions arise.

---

> > ### Comment · Reviewer_LrsF · 2025-08-06
> >
> > Thank you for the responses, which addressed most of my concerns. I will raise my score accordingly.

---

> > > ### Author Response · Authors · 2025-08-06
> > >
> > > Thank you for taking the time to review our paper and for recognizing our efforts. We sincerely appreciate your feedback and support!

---

### Official Review · Reviewer_31d8 · 2025-07-02

**Clarity:** 3
**Significance:** 2
**Originality:** 2
**Rating:** 4
**Confidence:** 3

**Summary:**

This paper proposes a modular GUI‐agent framework, InfantAgent-Next, that uses vision and tools (e.g., computer operations) to interact with GUI. The method uses UI-TARS-1.5-7B for visual grounding and Claude-3.7-Sonnet for planning. Its action space includes several predefined tools plus mouse‐click actions commonly used by pure‐vision agents. Results show that InfantAgent-Next was the best open‐source agent framework on OSWorld at the time.

**Questions:**

1. In terms of framework design, what are the main differences between your proposed framework and existing modular GUI agents such as MobileExperts, InfiGUIAgent, and AGENT S?
2. Have you tried using different backbone models?
3. I would like to see more examples like those in Figure 5, including both successful and failed cases, to better understand how the system works.

**Ethical Concerns:**

["NO or VERY MINOR ethics concerns only"]

**Final Justification:**

This paper proposes a new GUI-agent framework and, although it does not achieve state-of-the-art performance, has promising results. The framework integrates existing approaches without introducing significant novelty, but the experiments and case studies are comprehensive and convincing.

**Limitations:**

yes

**Quality:**

3

**Strengths And Weaknesses:**

Strengths

* Achieves competitive performance on the OSWorld, SWE, and GAIA leaderboards.
* Provides a detailed description of the proposed method.
* Conducts experiments across multiple GUI‐agent benchmarks.

Weakness
* The modular framework has little significant difference from existing modular GUI‐agent frameworks.

---

> ### Author Rebuttal · Authors · 2025-07-31
>
> Thank you for your positive assessment of our work! Below, we provide point‑by‑point responses to the comments.
> ***********
> **For weakness 1 and question 1**
> > In terms of framework design, what are the main differences between your proposed framework and existing modular GUI agents such as MobileExperts, InfiGUIAgent, and AGENT S?
>
> InfantAgent‑Next is a **hybrid framework that unifies tool‑based and pure‑vision agents**. Rather than relying solely on mouse/keyboard GUI manipulation, the agent exposes a broader action space under a unified interface—issuing shell commands, executing Python code, and performing mouse/keyboard actions. A planner chooses among these paths and can fall back to non‑visual execution when GUI grounding is uncertain. This design makes the system applicable to more complex workflows.
>
> For example, when the agent is asked to process a CSV file, a pure GUI agent would require a long sequence of clicks and drags, where a single grounding error can derail the task; by contrast, InfantAgent‑Next can simply write and run a short Python script to transform and process the CSV file, substantially improving robustness and end‑to‑end success.
>
> **For weakness 2**
> > Have you tried using different backbone models?
>
> We evaluated three backbone models: Claude‑3.7‑Sonnet, GPT‑4o, and DeepSeek‑V3‑0324. Because DeepSeek‑V3‑0324 lacks multimodal ability, we used it exclusively on the SWE‑Bench test. Since agent performance is strongly coupled with the backbone model’s capability, in our main comparisons against SOTA agents we report results using the strongest backbone per setting.
>
> Below we provide our measured results for GPT‑4o and DeepSeek‑V3‑0324:
>
> Table 1: SWE‑bench‑Verified test results.
> | Agent            | Model            | Accuracy |
> |:----------------:|:----------------:|:-------:|
> | OpenHands        | DeepSeek-V3-0324 |    $38.8\\%$ |
> | Agentless        | DeepSeek-V3-0324 |    $35.8\\%$ |
> | AutoCodeRover    | DeepSeek-V3-0324 |    $34.8\\%$ |
> | **InfantAgent-Next** | DeepSeek-V3-0324 |    $37.2\\%$ |
> | OpenHands        | GPT-4o           |   $41.20\\%$ |
> | Agentless        | GPT-4o           |   $38.80\\%$ |
> | AutoCodeRover    | GPT-4o           |   $39.20\\%$ |
> | **InfantAgent-Next** | GPT-4o           |   $40.20\\%$ |
>
> Table 2: OSWorld test results using GPT‑4o as the backbone model (maximum 50 steps per task).
>
> | Agent            | Accuracy |
> |:----------------:|:-------:|
> | Agent S          |   $20.58\\%$ |
> | OpenAI CUA       |    $32.6\\%$ |
> | Agent S2         |   $27.91\\%$ |
> | Aguvis           |   $17.04\\%$ |
> | **InfantAgent-Next** |   $30.35\\%$ |
>
> **For weakness 3**
> > I would like to see more examples like those in **Fig. 5**, including both successful and failed cases, to better understand how the system works.
>
> Here, we present two additional examples to detail the problem‑solving workflow of InfantAgent‑Next:
>
> Legend: italic text: agent‑generated output; ⚠️:annotation for ease of understanding.
>
> ************
>
> ***1. Positive example from GAIA (task ID: e1fc63a2-da7a-432f-be78-7c4a95598703).***
>
> Question: If Eliud Kipchoge could maintain his record-making marathon pace indefinitely, how many thousand hours would it take him to run the distance between the Earth and the Moon its closest approach? Please use the minimum perigee value on the Wikipedia page for the Moon when carrying out your calculation. Round your result to the nearest 1000 hours and do not use any comma separators if necessary.
>
> ****** **Agent Start to Work** ******
>
> **1. Assign task 1:**
>
> *Thought: First, I need to find the minimum perigee distance between Earth and Moon from Wikipedia.*
>
> *Task: Search on Wikipedia for the Moon page and find the minimum perigee (closest approach) distance between Earth and Moon.*
>
> **2. Start to finish task 1**
>
> *Tool Selection: browser* ⚠️ **select the suitable tools from toolkits.**
>
> *Thought: I'll navigate to the Wikipedia page for the Moon to find information.*
>
> *Action: open_browser(), navigate_to(en.wikipedia.org/wiki/Moon)*
>
> *Observation: Screenshot saved at /workspace/screenshots/1744924507.png*  ⚠️ **Agent can see this screenshot.**
>
> **3.  Assign task 2:**
>
> *Thought: Now I need to find Eliud Kipchoge's record marathon pace and calculate how long it would take him to run the distance to the Moon at that pace.*
>
> *Task: Search related information*
>
> **4. Start to finish task 2**
>
> *Tool Selection: browser*
>
> *Thought: I’ll search for Eliud Kipchoge's marathon world record time and pace.*
>
> *Action: google_search(\"Eliud Kipchoge marathon world record pace\")*
>
> *Observation: Screenshot saved at /workspace/screenshots/1744924600.png*
>
> **5. Assign task 3:**
>
> *Task: Calculate the final answer*
>
> **6. Start to finish task 3:**
>
> *Tool Selection: python*
>
> *Thought:  Now I have all the information, Let me perform the calculation*
>
> *Action: # python code to calculate the final answer, too long, omitted here.*
>
> *Observation: Final Answer: 17* ⚠️ **python code results.**
>
> **7. Provide final answer:**
>
> *Based on our calculations, I can provide the final answer in the requested format. FINAL ANSWER: 17*
>
> ************
>
> ***2. Negative example from SWE‑bench (task ID: astropy_astropy-7606).***
>
> Question:
> Unit equality comparison with None raises TypeError for UnrecognizedUnit
> ```
> In [12]: x = u.Unit('asdf', parse_strict='silent')
>
> In [13]: x == None  # Should be False
> ---------------------------------------------------------------------------
> TypeError                                 Traceback (most recent call last)
> <ipython-input-13-2486f2ccf928> in <module>()
> ----> 1 x == None  # Should be False
>
> /Users/aldcroft/anaconda3/lib/python3.5/site-packages/astropy/units/core.py in __eq__(self, other)
>   1699
>   1700     def __eq__(self, other):
> -> 1701         other = Unit(other, parse_strict='silent')
>   1702         return isinstance(other, UnrecognizedUnit) and self.name == other.name
>   1703
>
> /Users/aldcroft/anaconda3/lib/python3.5/site-packages/astropy/units/core.py in __call__(self, s, represents, format, namespace, doc, parse_strict)
>   1808
>   1809         elif s is None:
> -> 1810             raise TypeError("None is not a valid Unit")
>   1811
>   1812         else:
>
> TypeError: None is not a valid Unit
> ```
>
> ****** **Agent Start to Work** ******
>
> **1. Assign task 1:  ✔️**
>
> *Thought: It appears that the file "core.py" used in the traceback was found under a directory path that includes "units"*
>
> *Task: Locate the core.py file mentioned in the error traceback to understand the issue*
>
> **2. Start to finish task 1: ✔️**
>
> *Tool Selection: Bash*
>
> *Thought: Let's execute the Bash commands to display the content around "None is not a valid Unit" in core.py:*
>
> *Action: # Bash commands.*
>
> *Observation: # Bash commands results*
>
> **3. Assign task 2: ✔️**
>
> *Thought: We should first reproduce the issue.*
>
> *Task: Create a test script to reproduce the issue with UnrecognizedUnit equality comparison with None.*
>
> **4. Start to finish task 2: ✔️**
>
> *Tool Selection: Bash*
>
> *Thought: Based on the repository exploration, I'll create a minimal script that demonstrates the problem.*
>
> *Action: # Bash commands to write the code into the file and run the code.*
>
> *Observation: # Test script results.*
>
> **5. Assign task 3: ✔️**
>
> *Thought: We should implement the fix for the issue where UnrecognizedUnit equality comparison with None raises TypeError.*
>
> *Task: We need to modify the `__eq__` method in the UnrecognizedUnit class to handle None as a special case.*
>
> **6. Start to finish task 3: ❌**
>
> *Tool Selection: Bash*
>
> *Thought: Let's apply the sed command to insert the special case for None into the `__eq__` method:*
>
> *Action: # Bash commands to modify the code.*  ⚠️ **The modification does not handle certain edge cases.**
>
> *Observation: # Modified code*
>
> **7. Assign task 4: ✔️**
>
> *Thought: We need to verify that the code we modified is correct.*
>
> *Task: Let’s run the test script again.*
>
> **8. Start to finish task 4: ✔️**
>
> *Tool Selection: Bash*
>
> *Action: # Bash commands to run the code*
>
> *Observation: #  Test script results.*
>
> **9. Provide final answer: ✔️**
>
> *ANSWER: All tests have passed successfully, confirming that our fix resolves the issue without affecting any existing functionality in the test suite. Please check `/workspace` to see the patch.*
>
> ***********
> We once again thank the reviewer for the thoughtful comments. We hope our responses address your concerns, and we would be glad to continue the discussion should any further questions arise.

---

> > ### Comment · Reviewer_31d8 · 2025-08-04
> >
> > Thank you for the response. The cases are useful for readers and please put them in your paper. I will keep my score.

---

> > > ### Author Response · Authors · 2025-08-04
> > >
> > > Thank you for your reply! We will add these examples to the paper’s supplementary material.

---

### Official Review · Reviewer_c5md · 2025-07-03

**Clarity:** 2
**Significance:** 2
**Originality:** 2
**Rating:** 4
**Confidence:** 1

**Summary:**

The paper introduces InfantAgent-Next, a modular multimodal agent that performs real-world computer tasks by combining tool-based and vision-based approaches. It separates the agent workflow into three stages—planning, tool selection, and execution—each handled by specialized models. The system supports text, image, audio, and video inputs and can interact directly with GUIs. Key contributions include a unified framework for multimodal interaction, an iterative visual grounding method for accurate mouse clicks, and a robust file editing strategy.

**Questions:**

See weakness.

**Ethical Concerns:**

["NO or VERY MINOR ethics concerns only"]

**Final Justification:**

I’m not familiar with agent design, so I can not provide meaningful feedback. But after reviewing the other reviewers’ comments and the authors’ rebuttal, I recognize the value of the paper’s contribution.

**Limitations:**

Yes

**Quality:**

2

**Strengths And Weaknesses:**

**Strengths:**
1. Experiments cover a wide range of tasks and benchmarks, showing the agent's effectiveness.
2. Practical improvements such as iterative visual grounding and robust file editing effectively mitigate common failure modes in current agent systems.

**Weaknesses:**
1. The core ideas seems mostly integration of known components rather than novel contributions.
2. The iterative visual grounding method for mouse clicking is intuitive and effective in narrowing down regions. However, it assumes that the initial grounding is at least roughly correct. If the initial prediction is wrong, later iterations may focus on irrelevant regions. The paper does not discuss whether the agent can detect failure (e.g., no object found) and roll back or restart from a broader view.

---

> ### Author Rebuttal · Authors · 2025-07-31
>
> We thank the reviewer for the thoughtful and constructive feedback. Below, we provide point‑by‑point responses to the comments.
> ***********
> **For weakness 1**
> > The core ideas seem mostly integration of known components rather than novel contributions.
>
> **1. Our contributions:**
>
> Based on the NeurIPS 2025 Call for Papers taxonomy, we position InfantAgent-Next in the **Infrastructure category** (e.g., libraries, improvements in implementation and scalability, and distributed solutions). We have two main contributions:
>
> - Infrastructure for computer-use agents. **We present an infrastructure for building multimodal, general-purpose agents with integrated computer-use capability**—i.e., agents that can operate their own computers.
>
>
> - Implementation improvements via unification and interfaces. **We unify tool-based and purely vision-based approaches into a single stack and expose unified interfaces that broaden agents’ computer-use capabilities**. We hope this serves as a versatile reference for the community.
>
> **2. Our novel methods for new challenges**
>
> When integrating pure-vision agents with tool‑based agents to build multimodal, general‑purpose agents, we encounter several categories of challenges not present in prior single‑paradigm agents. Below, we describe each challenge in turn and present our novel solutions.
> ***********
> ***Challenge 1*** — Growth in tool count inflates inference cost.
>
> Traditional tools‑based and pure‑vision agents each expose their own action space. For example, OpenHands (tool‑based) offers file‑editing commands such as `file_edit` and `file_open`, whereas UI‑TARS (pure‑vision) provides mouse/keyboard primitives like `mouse_click` and `press_key`. When these paradigms are fused, we must unify all action spaces and, to handle additional modalities (e.g., audio/video), introduce extra tools. All of these tools—together with usage examples—need to be registered at the start of the dialogue (e.g., in the tool manifest/system prompt), which substantially increases context length and token usage, thereby raising inference cost and latency relative to single‑paradigm agents.
>
> ***Our novel method for challenge 1***:
> We introduce a tool‑selection step  (**Fig. 2** and Lines 109–118): after the planner produces a subtask plan, the agent performs task‑conditioned tool selection, activating only the minimal set of tools needed for the upcoming subtask.
>
> Original workflow:
> ```
> agent.register_tools()        # registers/activates everything up front
> await agent.planning()
> await agent.execution()
> ```
> Our workflow:
> ```
> await self.planning()
> await self.tool_selection() # selects a minimal tool set for this subtask
> await self.execution()
> ```
>
> ***Example***
> As illustrated in **Fig. 5**, the three tool‑selection phases (blue shaded regions) choose different categories of tools conditioned on the current subtask, rather than registering the entire tool inventory to the model. This selective exposure reduces inference cost.
> ***********
> ***Challenge 2*** — Rising task complexity vs. limited model capacity.
>
> Traditional tool‑based agents typically rely on a single backbone model to handle an entire task—AutoGPT and BabyAGI, for instance, route planning and execution through one (or at most a two‑slot fast/slow) LLM—whereas some vision‑based agents augment the main model with a dedicated GUI visual‑grounding component for pointer‑level localization, as in OS‑Atlas. However, real‑world multimodal agents must operate across text, images, audio, and video. Given this breadth and cross‑domain complexity, a single model—or a small fixed set of models—is often insufficient to deliver accurate and reliable performance.
>
> ***Our novel method for challenge 2***:
> We leverage the agent’s modular design: the end-to-end workflow is decomposed into well-defined modules, and subtasks are routed to specialized models. Difficult, cross-domain steps are assigned to large, high-capability backbones—e.g., we rely on Claude-3.7-Sonnet for global task planning. We assign latency-sensitive operations to lower-latency models; e.g., Qwen2.5-72B-Instruct is used for precise file-editing commands. Vision subtasks go to vision models, and audio subtasks to audio models. This task‑conditioned model routing preserves accuracy while reducing cost and latency. Implementation details are provided in **Fig. 2** and Lines 101–108.
>
> ***Example***. In **Fig. 5**, the execution step beneath Turn 1 (purple region) corresponds to image analysis performed by a vision model, whereas the execution steps under Turns 2–3 are carried out by a code‑specialized model.
> This setup exploits complementary strengths and reduces failures due to modality limits or capability shortfalls.
>
> ***********
> ***Challenge 3*** — Tool‑use hallucination
>
> Even SOTA models can hallucinate when invoking tools that require precise numeric arguments. Typical failure modes include providing an incorrect line number to the file‑editing tool or wrong coordinates to the visual‑grounding tool. For example, in our tests reported in **Sec. 4.5**, DeepSeek‑V3‑0324 shows a $61\\%$ tool‑use hallucination rate on file‑editing commands without any mitigation; in long‑horizon multimodal tasks, such small errors compound and can ultimately lead to end‑to‑end failure.
>
> ***Our novel method for challenge 3***: We design two algorithms tailored to these error modes that substantially reduce tool‑use hallucinations.
>
> **Alg. 1** (Iterative Region Cropping & Mouse Click): Repeatedly ground → crop → re-ground to progressively narrow the region before the final click, reducing misclicks without retraining and working with any GUI grounder.
>
> **Alg. 2** (File Editing: Plan-Verify-Repair): Have the model propose edit boundaries with content anchors, verify them against the file, and on mismatched fuzzy re-localize and repair before applying the edit, curbing line-number hallucinations.
>
> see **Alg. 1** and **Alg. 2** for details.
>
> ***Effectiveness of method 3***. In our ablation studies (**Sec. 4.4** and **Sec. 4.5**), **Alg. 1** increases visual‑grounding accuracy on high‑resolution images by $11.6$ percentage points, while **Alg. 2** corrects $84.3\\%$ of file‑editing tool sequencing errors.
>
> **For weakness 2**
>
> > The iterative visual grounding method for mouse clicking is intuitive and effective in narrowing down regions. However, it assumes that the initial grounding is at least roughly correct. If the initial prediction is wrong, later iterations may focus on irrelevant regions. The paper does not discuss whether the agent can detect failure (e.g., no object found) and roll back or restart from a broader view.
>
>
> In practice, initial mis-grounding that could steer later iterations to irrelevant regions is rare: on ScreenSpot‑V2 (a standard‑resolution benchmark), our default GUI‑grounding model UI‑TARS‑1.5‑7B attains $88.7\\%$ accuracy; adding our iterative narrowing raises it to $94.6\\%$. Beyond accuracy, InfantAgent‑Next integrates tool‑based and vision‑only capabilities, so when GUI grounding is uncertain the agent can fall back to non‑visual action paths (e.g., terminal commands or Python scripts) to complete the task.
>
> Here is an example from OSWorld benchmark (id: 3aaa4e37-dc91-482e-99af-132a612d40f3):
>
> Question: *Could you help me to export the current sheet to a csv file? Export the contents just as they are shown on the screen. Just keep the other options untouched. A default csv format is ok. The csv should share the file name with the original xlsx.*
>
> Agent's actions: First, the agent searches for the `Save As` button and the `CSV` format. Finding no `CSV` option on screen, it switches strategy and executes an one‑line Bash command, which resolves the question.
> ***********
> We once again thank the reviewer for the thoughtful comments. We hope our responses address your concerns, and we would be glad to continue the discussion should any further questions arise.

---

> > ### Comment · Reviewer_c5md · 2025-08-04
> >
> > Thank you for your detailed reply. I’m sorry, but I’m not familiar with agent design, so I can not provide meaningful feedback. After reviewing the other reviewers’ comments and the authors’ rebuttal, I recognize the value of the paper’s contribution and am willing to raise my score to 4.

---

> > > ### Author Response · Authors · 2025-08-04
> > >
> > > We appreciate your response and your supportive assessment of our work!

---

### Note · Authors · 2025-08-12

We thank the reviewers and AC for all the efforts in reviewing our paper and providing helpful and insightful feedback! Below we summarize the rebuttal content to facilitate the final decision.

*****

1. We are happy to see that all reviewers maintained a positive attitude toward our work after the rebuttal, especially recognizing the following points:

- Our work is highly significant for the field of automated AI agents, addressing practical challenges in multimodal human-computer interaction.

- Our paper is clearly written and provides a detailed description of the proposed method as well as in-depth case analyses.

- Our agent has been thoroughly evaluated on multiple real-world datasets.


2. During the rebuttal process, we provided detailed responses to the reviewers’ questions, with the key points summarized as follows:

- Elaborated on our contributions.

- Supplemented the experimental results using different backbone models.

- Added positive and negative example cases.

3. The reviewers provided positive feedback:

- Reviewer x34C gave a score of 5 with a confidence level of 5, and we are deeply grateful for the recognition of our work!

- Reviewers LrsF and c5md indicated that our responses addressed their concerns and decided to raise their scores.

- Reviewer 31d8 chose to maintain the positive evaluation of our work.

*****

We would like to once again thank all the reviewers and the AC for their contributions during the review process, and we hope that our work can contribute to the advancement of the relevant field.

---

### Decision · Program_Chairs · 2025-09-17

**Decision:**

Accept (poster)

**Comment:**

The paper introduces InfantAgent-Next, a modular multimodal agent integrating tool-based and vision-based approaches for real-world computer use, with strong results on OSWorld, SWE-Bench, and GAIA. Strengths: clear writing, thorough experiments, practical engineering (tool selection, modular routing, hallucination reduction), and open-source release. Weaknesses: limited novelty, integration of known methods, some fairness and scalability concerns. Despite modest originality, the work makes solid infrastructure contributions with strong empirical value. In rebuttal, authors clarified contributions, added ablations and examples, and addressed fairness/cost concerns; most reviewers raised scores, while one kept accept but noted limited novelty.